# Offline Equilibrium Finding

## Abstract

Offline reinforcement learning (offline RL) is an emerging field that has recently attracted significant interest across a wide range of application domains, owing to its ability to learn policies from previously collected datasets. The success of offline RL has paved the way for tackling previously intractable real-world problems, but so far, only in single-agent scenarios. Given its potential, our goal is to generalize this paradigm to the multiplayer-game setting. To this end, we introduce a novel problem, called *offline equilibrium finding* (OEF), and construct various types of datasets spanning a wide range of games using several established methods. To solve the OEF problem, we design a model-based framework capable of directly adapting any online equilibrium finding algorithm to the OEF setting while making minimal changes. We adapt the three most prominent contemporary online equilibrium finding algorithms to the context of OEF, resulting in three model-based variants: OEF-PSRO and OEF-CFR, which generalize the widely-used algorithms PSRO and Deep CFR for computing Nash equilibria, and OEF-JPSRO, which generalizes the JPSRO for calculating (coarse) correlated equilibria. Additionally, we combine the behavior cloning policy with the model-based policy to enhance performance and provide a theoretical guarantee regarding the quality of the solution obtained. Extensive experimental results demonstrate the superiority of our approach over traditional offline RL algorithms and highlight the importance of using model-based methods for OEF problems. We hope that our work will contribute to the advancement of research in large-scale equilibrium finding.

## 1 Introduction

Game theory provides a universal framework for modeling interactions among cooperative and competitive players (Shoham & Leyton-Brown, 2008). The canonical solution concept is Nash equilibrium (NE), describing a situation when no player increases their utility by unilaterally deviating. However, computing NE in two-player or multi-player general-sum games is PPAD-complete (Daskalakis et al., 2006; Chen & Deng, 2006), which makes solving games, whether exactly or approximately, exceedingly difficult. The complexity persists even in two-player zero-sum games, regardless of whether the players can perceive the game state perfectly (e.g., in Go (Silver et al., 2016)) or imperfectly (e.g., in poker (Brown & Sandholm, 2018) or Star-Craft II (Vinyals et al., 2019)). In recent years, learning algorithms have demonstrated their superiority over traditional optimization methods, such as linear or nonlinear programs in solving large-scale imperfect-information extensive-form games. Particularly, the most successful learning algorithms belong either to the line of research on counterfactual regret minimization (CFR) (Brown & Sandholm, 2018), or policy space response oracles (PSRO) (Lanctot et al., 2017). CFR is an iterative algorithm that approximates NEs through repeated self-play. Several sampling-based CFR variants (Lanctot et al., 2009; Gibson et al., 2012) have been proposed to efficiently solve large games. To further scale up to even larger games, CFR can be embedded with neural network function approximation (Brown et al., 2019; Steinberger, 2019; Li et al., 2019; Agarwal et al., 2020). The other algorithm, PSRO, generalizes the double oracle method (McMahan et al., 2003; Bošanský et al., 2014) by incorporating (deep) reinforcement learning (RL) methods as a best-response oracle (Lanctot et al., 2017; Muller et al., 2019). Especially, neural fictitious self-play (NFSP) can be considered as a special case of PSRO (Heinrich et al., 2015). Both CFR and PSRO have achieved impressive performance, particularly in the more challenging realm of large-scale imperfect-information extensive-form games such as poker (Brown & Sandholm, 2018; McAleer et al., 2020).

A critical component contributing to the success of these learning algorithms is the availability of *efficient and accurate simulators.* Simulators can be constructed using rules, such as in various poker games (Lanctot et al., 2019), or using a video-game suit like StarCraft II (Vinyals et al., 2017). These simulators serve as environments, enabling agents to collect millions or even billions of trajectories to facilitate the training process. We can consider this mode of learning, which relies on a simulator, as an online mode since it can access data from the simulator at any time. In many real-world games, such as football (Kurach et al., 2020; Tuyls et al., 2021) or table tennis (Ji et al., 2021), learning in online mode is not practical, as constructing a sufficiently accurate simulator may be infeasible due to numerous complex factors affecting the game-play. These factors include relevant laws of physics, environmental conditions (e.g., wind speed), or physiological limits of (human) bodies rendering certain actions unattainable. Consequently, football teams or table tennis players may resort to watching previous matches to improve their strategies, which semantically corresponds to the offline learning mode, i.e., learning from previously collected data. In recent years, there have been several (often domain-specific) attempts to formalize offline learning in the context of games. For instance, StarCraft II Unplugged (Mathieu et al., 2021) provides a dataset of human game-plays in this two-player zero-sum symmetric game. Concurrently, some works (Cui & Du, 2022; Zhong et al., 2022) explore the necessary properties of offline datasets of two-player zero-sum Markov games to successfully infer their NEs.

However, these prior works mainly focus on solving Markov games, while our goal is to solve extensive-form games in the offline setting. Detailly, Cui & Du (2022) focus on computing the Nash equilibrium strategy for tabular two-player zero-sum Markov games, while our work not only focuses on computing the NE strategy for two-player zero-sum extensive-form games but also focuses on computing the C(CE) for multi-player extensive-form games. Furthermore, their theoretical results are focused on the two-player zero-sum Markov game while our work extends their results to the extensive-form game setting. To our understanding, there has been no comprehensive study focusing on multi-player games in an offline setting. Moreover, there is a notable absence of systematic definitions and research efforts aimed at formalizing offline learning within the context of games. To address this gap, we put forward a more general problem called *offline equilibrium finding* (OEF), which aims to identify the equilibrium strategy of the underlying game based on a fixed offline dataset collected by an unknown behavior strategy. The lack of an accurate simulator in offline settings complicates the process of identifying equilibrium strategies, as it makes evaluating and validating learned strategies more difficult. Consequently, the OEF problem poses a significant challenge, as it necessitates forging a connection between an equilibrium strategy and an offline dataset. To tackle this problem, we introduce an environment model that serves as an intermediary between the equilibrium strategy and the offline dataset. Our contributions can be summarized as follows:

- We introduce a new paradigm, Offline Equilibrium Finding (OEF), highlighting the challenges associated with learning equilibrium strategies only from offline datasets without an accurate simulator.

- We create OEF datasets from widely recognized game domains to better define the OEF problem. To achieve this, we employ various methods to produce different behavior strategies for generating offline data, enabling our OEF datasets to encompass a diverse range of potential gameplay scenarios.

- We propose a novel OEF algorithm, BCMB, that combines a simple model-free algorithm (behavior cloning technique) with an innovative model-based framework. This model-based framework has the capability to generalize any online equilibrium finding algorithm to the OEF setting by introducing an environment model as an intermediary. Furthermore, we investigate the relationship between the data coverage of the offline dataset and the performance of the offline algorithm, and we provide a guarantee of the solution quality for our OEF algorithm.

- We conduct comprehensive experiments to evaluate the effectiveness of our proposed OEF algorithm. The experimental results substantiate the superiority of our algorithm over model-based and model-free offline RL algorithms and the efficiency of our algorithm for solving the OEF problem.

We hope our work can provide a broader understanding of offline learning in multi-player games and establish a foundation for future research in this emerging area.

## 2 Preliminaries

In this section, we first introduce the imperfect-information extensive-form game model focused on in this paper, and then we introduce two types of widely-used equilibrium-finding algorithms, namely Counterfactual Regret Minimization (CFR) and Policy Space Response Oracles (PSRO).

### 2.1 Imperfect-Information Extensive-form Games

An imperfect-information extensive-form game (Shoham & Leyton-Brown, 2008) can be represented as a tuple $(N, H, A, P, \mathcal{I}, u)$, where $N = \{1, ..., n\}$ is a set of players, $H$ is a set of histories (i.e., the possible action sequences) and $A$ is a set of actions available to each player. The empty sequence $\emptyset$ corresponds to a unique root node of a game tree, which is included in $H$. Additionally, every prefix of a sequence in $H$ is also in $H$. A special subset of the set of histories is $Z \subset H$ which corresponds to the set of terminal histories. $A(h) = \{a : (h, a) \in H\}$ is the set of actions available at any non-terminal history $h \in H \setminus Z$. $P$ is the player function, which maps each non-terminal history to a player, i.e., $P(h) \mapsto N \cup \{c\}, \forall h \in H \setminus Z$, where $c$ denotes the "chance player", which represents stochastic events outside of the players' controls. In other words, $P(h)$ is the player who takes an action at the history $h$. If $P(h) = c$ then chance determines the action taken at history $h$. $\mathcal{I}$ denotes the set of the information set. The information set $\mathcal{I}_i$ forms a partition over the set of histories where player $i \in N$ takes action, such that player $i \in \mathcal{N}$ cannot distinguish these histories within the same information set $I_i$. Therefore, every information set $I_i \in \mathcal{I}_i$ corresponds to one decision point of player $i$ which means that $P(h_1) = P(h_2)$ and $A(h_1) = A(h_2)$ for any $h_1, h_2 \in I_i$. For convenience, we use $A(I_i)$ to represent the action set $A(h)$ and $P(I_i)$ to represent the player $P(h)$ for any $h \in I_i$. For $i \in N$, a utility function $u_i : Z \to \mathbb{R}$ specifies the payoff of player $i$ for every terminal history.

The behavior strategy of player $i$, $\sigma_i$, is a function mapping every information set of player $i$ to a probability distribution over $A(I_i)$, and $\Sigma_i$ is the set of strategies for player $i$. A strategy profile $\sigma$ is a tuple of strategies, one for each player, $(\sigma_1, \sigma_2, ..., \sigma_n)$, with $\sigma_{-i}$ referring to all the strategies in $\sigma$ except $\sigma_i$. Let $\pi^\sigma(h) = \prod_{i \in N \cup \{c\}} \pi_i^\sigma(h)$ be the reaching probability of history $h$ when all players choose actions according to $\sigma$, where $\pi_i^\sigma(h)$ is the contribution of $i$ to this probability. Given a strategy profile $\sigma$, the expected value to player $i$ is the sum of expected payoffs of these resulting terminal nodes, $u_i(\sigma) = \sum_{z \in Z} \pi^\sigma(z) u_i(z)$.

The canonical solution concept for imperfect information extensive form games is Nash equilibrium (NE), where no player can increase their expected utility by unilaterally switching to a different strategy. Formally, the strategy profile $\sigma^*$ forms an NE if it satisfies $u_i(\sigma^*) = \max_{\sigma'_i \in \Sigma_i} u_i(\sigma'_i, \sigma^*_{-i}) \geq u_i(\sigma_i, \sigma^*_{-i}), \quad \forall \sigma_i \in \Sigma_i, i \in N$. To quantify the distance between a strategy profile $\sigma$ and the NE strategy, we introduce the metric $\text{NASHCONV}(\sigma) = \sum_{i \in N} \text{NASHCONV}_i(\sigma)$, where $\text{NASHCONV}_i(\sigma) = \max_{\sigma'_i} u_i(\sigma'_i, \sigma_{-i}) - u_i(\sigma)$ for each player $i$. Especially, for two-player zero-sum games, the metric simplifies to $\text{NASHCONV}(\sigma) = \sum_{i \in N} \max_{\sigma'_i} u_i(\sigma'_i, \sigma_{-i})$. When $\text{NASHCONV}(\sigma) = 0$, it indicates that $\sigma$ is the NE strategy. However, for $n$-player general-sum games, apart from NE, (Coarse) Correlated Equilibrium ((C)CE) is more commonly employed as the solution concept. Similar to the NE strategy, a Correlated Equilibrium (CE) strategy is a joint mixed strategy in which no player has the incentive to deviate (Moulin & Vial, 1978). Formally, let $S_i$ represent the strategy space for player $i$ and $S$ represent the joint strategy space. The strategy profile $\sigma^*$ forms a CCE if it satisfies for $\forall i \in N, s_i \in S_i, u_i(\sigma^*) \geq u_i(s_i, \sigma^*_{-i})$ where $\sigma^*_{-i}$ is the marginal distribution of $\sigma^*$ on strategy space $S_{-i}$. Analogous to NE, the (C)CE Gap Sum is adopted to measure the gap between a joint strategy and the (C)CE (Marris et al., 2021).

### 2.2 Equilibrium Finding Algorithms

**PSRO.** The Policy Space Response Oracles (PSRO) algorithm (Lanctot et al., 2017) begins with an initial set of randomly-generated policies, $\hat{\Sigma}_i$ for each player $i$. During each iteration of PSRO, a meta-game $M$ is built using all existing policies of the players by simulation. A meta-solver then computes a meta-strategy, which is a distribution over the policies of each player (e.g., Nash, $\alpha$-rank, or uniform distributions). The joint meta-strategy for all players is represented as $\alpha$, where $\alpha_i(\sigma)$ denotes the probability that player $i$ selects $\sigma$ as their strategy. Subsequently, an oracle computes at least one best response policy for each player, which is then added to $\hat{\Sigma}_i$. It is important to note when computing a new policy for a player, the

policies of all other players and the meta-strategy remain fixed. This leads to a single-player optimization problem that can be solved by DQN (Mnih et al., 2015) or policy gradient reinforcement learning algorithms. Neural Fictitious Self-Play (NFSP) can be considered as a special case of PSRO that employs the uniform distribution as the meta-strategy (Heinrich et al., 2015). Joint Policy Space Response Oracles (JPSRO) is an innovative extension of PSRO that incorporates fully mixed joint policies to facilitate coordination among policies (Marris et al., 2021). JPSRO is proven to converge to a (C)CE over joint policies in extensive-form games. The details for the process of PSRO can be found in Lanctot et al. (2017) and we also provide the pseudocode for MB-PSRO in Appendix E which is similar to PSRO.

**CFR.** Counterfactual Regret Minimization (CFR) (Zinkevich et al., 2007) is an iterative algorithm for approximately solving large imperfect-information games. In each iteration, the entire game tree is traversed, and the counterfactual regret for every action $a$ in every information set $I$ is computed. The computation of the counterfactual regret value for a player's information set is associated with the counterfactual value of the information set, which is the expected value of the information set given that the player tries to reach it. After traversing the game tree, players employ *Regret Matching* to select a distribution over actions in every information set, proportional to the positive cumulative regret values of those actions. In the next iteration, players use the new strategy to traverse the entire game tree and this process is repeated until convergence. In two-player zero-sum games, if the average regret for both players is less than $\epsilon$, their average strategies over strategies in all iterations $(\overline{\sigma}_1^T, \overline{\sigma}_2^T)$ form a $2\epsilon$-equilibrium (Waugh et al., 2009). The details for the CFR algorithm can be found in Zinkevich et al. (2007) and we also provide the detailed implementation for the MB-CFR algorithm which is similar to CFR in Appendix E. More recent studies have adopted deep neural networks to approximate counterfactual values, resulting in superior performance compared to their tabular counterparts (Brown et al., 2019; Steinberger, 2019; Li et al., 2019; 2021).

## 3 Problem Statement

To emphasize the importance of introducing the Offline Equilibrium Finding (OEF) problem, we will first present a motivating scenario that demonstrates the need for addressing the OEF problem. Following this, we will explain the limitations of current algorithms in solving the OEF problem. Lastly, we will introduce the OEF problem itself, along with the challenges it poses.

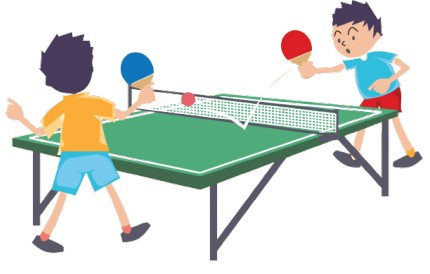

(a) The game of table tennis.

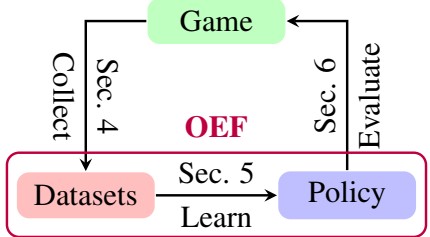

(b) The illustration of OEF problem

Figure 1: The example and illustration of OEF problem

**Motivating Scenario.** Assume that a table tennis player $A$ is preparing to compete against player $B$, whom they never faced before (Figure 1(a)). In this situation, what could player $A$ do to prepare for the match? Although player $A$ understands the rules of table tennis, they lack specific knowledge about playing against player $B$, such as their preferred moves or actions and their subjective payoff function. Without this detailed game information, player $A$ cannot build an accurate simulator to simulate the game they will play, rendering self-play or other online equilibrium-finding algorithms ineffective. Moreover, if player $A$ simply adopts the best response strategy against player $B$'s previous strategy, this approach may be exploited if player $B$ changes their strategy. Consequently, player $A$ must *watch the matches that player $B$ played against other players* to learn their style and compute the equilibrium strategy, which minimizes exploitation, of the

underlying game they will play. This process aligns with the proposed OEF methodology. Next, we will present the definition of the OEF problem and discuss the challenges associated with it.

**Offline Equilibrium Finding.** Based on the motivating scenario, we observe that in games with complex dynamics, such as table tennis games or football games (Kurach et al., 2020), it is difficult to build a realistic simulator or learn the policy during playing the game. An alternative solution is to learn the policy from the historical game data. To characterize this situation, we propose the *offline equilibrium finding* (OEF) problem: *Given a fixed dataset $\mathcal{D}$ collected by an unknown behavior strategy $\sigma$, find an equilibrium strategy profile $\sigma^*$ of the underlying game.*

In order to gain a deeper understanding of the OEF problem, we will illustrate the OEF problem using Figure 1(b) and provide a formal definition of the offline equilibrium finding problem as follows,

**Definition 3.1** (OEF). Given a game's offline dataset $\mathcal{D} = (s_t, a, s_{t+1}, r_{t+1})$ where $s_t$ and $s_{t+1}$ refer to the game states, $a$ refers to the action played at $s_t$ and $r_{t+1}$ refers to the reward after performing action $a$ at $s_t$. The strategy used to collect the dataset $\mathcal{D}$ is unknown. The OEF problem is to find an approximate equilibrium strategy profile $\sigma^*$ that achieves a small gap between $\sigma^*$ and equilibrium, i.e., the NASHCONV for NE and (C)CE Gap Sum for (C)CE, only based on $\mathcal{D}$.

The OEF problem is similar to the Offline RL problem but poses several unique challenges: i) the canonical solution concept in the OEF problem is the equilibrium strategy, which necessitates an iterative procedure of computing best responses; ii) the game in the OEF problem involves at least two players competing against each other, which amplifies sensitivity to distribution shifts and other uncertainties compared to the Offline RL problem; and iii) the distribution shifts of opponents' actions and the dynamic of the game are coupled, complicating the process of distinguishing and addressing these issues. We list a comparison of OEF with these related works in Table 1. Further discussion about these related works can be found in Appendix B.

| Methods | Work w/o env | Converge to equilibrium |
|---|:---:|:---:|
| Offline RL (Lange et al., 2012; Levine et al., 2020) | ✓ | ✗ |
| Opponent Modelling (He et al., 2016) | ✗ | ✗ |
| Empirical Game-Theoretic Analysis (Wellman, 2006) | ✗ | ✓ |
| OEF | ✓ | ✓ |

Table 1: Comparison of OEF with other related methods.

## 4 Collection of Offline Datasets

As delineated in the OEF problem, a crucial element is an offline dataset of the game. Typically, this offline dataset is collected with unspecified strategies within real-world scenarios. Nevertheless, to effectively evaluate and analyze the performance of the offline algorithm in solving the OEF problem, a solitary dataset is insufficient. The reason is that the strategy employed to generate an offline dataset is unknown in the OEF problem, and relying on a single dataset for evaluation introduces bias. Consequently, an appropriate dataset benchmark for the OEF problem should consist of a diverse set of datasets that closely resemble real-world situations. Nonetheless, generating such a collection of offline datasets with high diversity presents a substantial challenge, as these offline datasets should be meaningful rather than merely comprising randomly generated datasets, even though such datasets may indeed display significant diversity. To mitigate this issue, we propose various methods for collecting a range of datasets that serve as the foundation for OEF research. These methods are all inspired by different real-world cases, ensuring that the resulting datasets are diverse and capable of mimicking real-world situations. We will now proceed to describe these collection methods.

### 4.1 Data Collection Methods

**Random Method.** When playing an unfamiliar game, the most common approach is initially exploring the game by making random actions. Our random method is inspired by this natural tendency for exploration

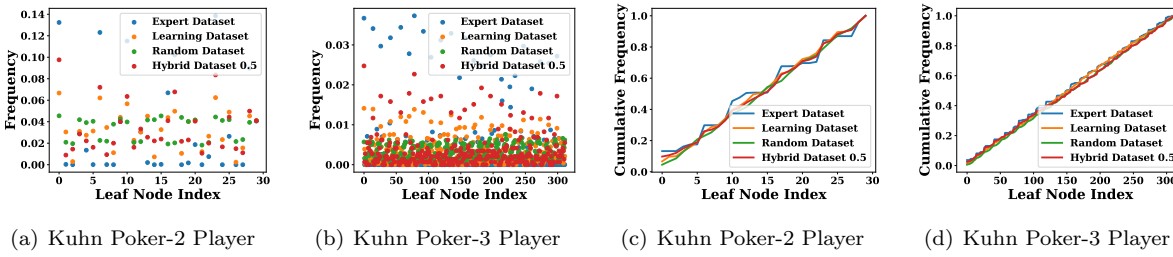

| (a) Kuhn Poker-2 Player | (b) Kuhn Poker-3 Player | (c) Kuhn Poker-2 Player | (d) Kuhn Poker-3 Player |

Figure 2: Dataset visualization

and simulates the experience of a beginner learning and familiarizing themselves with the game for the first time. The random method consists of three steps. In the first step, we assign a uniform strategy to each player in the game. During the second step, players repeatedly participate in the game. Finally, we collect the game data generated throughout the game-play. By following these steps, we obtain a dataset generated through the random strategy, which we refer to the **Random Dataset**.

**Learning Method.** Once players become familiar with the game, they tend to develop a non-exploitable strategy, such as the Nash equilibrium strategy, to improve their game-play. This observation inspires our learning method, which simulates the process of a player acquiring a non-exploitable strategy. To simulate the process of player learning, we employ the process of equilibrium finding algorithm. Therefore, our learning method used to collect datasets requires running one existing equilibrium-finding algorithm, such as CFR or PSRO. As the algorithm iterates through the equilibrium-finding process, we gather these intermediate interaction game data during each iteration and store it as the **Learning Dataset**.

**Expert Method.** The inspiration for this method comes from the notion that, when learning a game, we often benefit from observing more experienced players in action. We assume that the expert always employs a Nash equilibrium strategy, which is a non-exploitable strategy. Subsequently, we can adopt a similar methodology as in the random method. First, we assign the NE strategy, which can be computed using any existing equilibrium finding algorithm, to each player in the game. As a second step, these Nashian players repeatedly interact in the game. Finally, we gather the game data generated during game-play and store it as the **Expert Dataset**. However, in multi-player or general-sum games, computing the NE strategy is challenging. In this case, we can still employ the existing equilibrium finding algorithm to derive a suitable strategy. For instance, the PSRO algorithm with $\alpha$-rank as the meta-solver (Muller et al., 2019) can yield a fairly effective strategy (with low exploitability) in general-sum, multi-player games.

**Hybrid Method.** To simulate more realistic scenarios and generate a diverse range of datasets, we propose a hybrid method that combines the random dataset and the expert dataset in varying proportions. This approach enables the creation of a more comprehensive and diverse collection of datasets that better represent real-world situations. We refer to these combined datasets as **hybrid datasets**.

In this paper, we construct a dataset benchmark for the OEF problem by collecting data from player interactions in the most frequently used benchmark imperfect-information extensive-form games, which are prevalent in contemporary research on equilibrium finding. These games encompass poker games (two-player and multi-player Kuhn poker, two-player and multi-player Leduc poker), Phantom Tic-Tac-Toe, and Liar's Dice. The diverse datasets of these game data serve as the foundation for our OEF problem.

## 4.2 Visualizations of Collected Datasets

In accordance with the aforementioned collection methods, the collected datasets closely resemble real-world situations. To validate the diversity of these collected offline datasets and gain deeper insights into them, we introduce a visualization method for comparing them. Firstly, we generate the game tree for the corresponding game. Subsequently, we traverse the game tree using depth-first search (DFS) and assign an index to each leaf node based on the DFS results. Lastly, we count the frequency of each leaf node within the dataset. We focus solely on the frequency of leaf nodes because each leaf node represents a unique sampled trajectory originating from the root node of the game tree. As a result, the frequency of leaf nodes can

effectively capture the distribution of the dataset. To visualize and compare these offline datasets, a range of statistical methods can be employed on the collected frequency data of the leaf nodes.

The simplest methods for visualization involve plotting the frequency and cumulative frequency of leaf nodes. Figure 2 displays these datasets for two-player and three-player Kuhn games. From these figures, we can observe that in the random dataset, the frequency of leaf nodes is nearly uniform, whereas, in the expert dataset, the frequency distribution of leaf nodes is uneven. The distribution of the learning dataset and the hybrid dataset falls between that of the expert dataset and the random dataset. These observations confirm that the distributions of these datasets differ, thus validating the diversity of our proposed datasets. To provide more insight into our OEF datasets, we also apply other statistical methods, such as the Fourier transform. Additional visualization results can be found in Appendix C.

# 5 Algorithms for Offline Equilibrium Finding

Drawing inspiration from Offline RL (Chen et al., 2020; Yu et al., 2020), there are two possible approaches for solving the OEF problem: model-free and model-based approaches.

The model-free approach aims to learn a policy *directly* from the offline dataset, necessitating the establishment of a *direct* relationship between the equilibrium strategy and the offline dataset. The most straightforward method to achieve this is by applying the behavior cloning technique. However, the behavior cloning technique performs well only in certain cases. Specifically, if the offline dataset is generated using an equilibrium strategy, the behavior cloning technique can directly learn the equilibrium strategy from the offline dataset. However, the behavior cloning technique fails to produce satisfactory results when the strategy of the offline dataset is not an equilibrium strategy. Our experimental results also support this assertion. Moreover, considering that we cannot use the data of any two action tuples to determine which action tuple is closer to an equilibrium strategy, as equilibrium identification requires other action tuples to serve as references, other model-free algorithms are insufficient for solving the OEF problem since the model-free approach cannot measure the distance from the equilibrium strategy to guide the training process.

The model-based approach typically involves introducing a model to assist in learning an optimal strategy when addressing offline RL problems. Likewise, we can propose a model-based approach for tackling the OEF problem by incorporating an environment model as an intermediary between the equilibrium strategy and the offline dataset. However, our proposed model-based algorithm also cannot perform well in all cases, particularly when the offline dataset does not cover the majority of the game states. Our experimental results support this claim. As neither a single model-free nor model-based approach can perform well in all scenarios, we ultimately propose a novel algorithm – **BCMB**, which combines the model-free approach and the model-based approach for effectively solving the OEF problem. In the subsequent sections, we initially explain how the behavior cloning technique and the model-based algorithm work to solve the OEF problem. Following that, we introduce our OEF algorithm, the combination method: BCMB.

## 5.1 Behavior Cloning Technique

Behavior cloning (BC) is a method that imitates the behavior policy present in the dataset and is frequently used in solving offline RL (Fujimoto & Gu, 2021). In the OEF setting, we can also employ the BC technique to learn a behavior cloning strategy for every player from the offline dataset. More specifically, we can utilize the imitation learning algorithm to train a policy network $\sigma_i$, parameterized by $\theta$, for each player $i$ to predict the strategy for any given information set $I_i$. Only the information sets and action data are required when training the behavior cloning strategy. We use the cross-entropy loss as the training loss, defined as $\mathcal{L}_{bc} = \mathbb{E}_{(I_i,a) \sim D}[l(a, \sigma_i(I_i; \theta))] = -\mathbb{E}_{(I_i,a) \sim D}[a \cdot \log(\sigma_i(I_i; \theta))]$, where $a$ represents the one-hot encoding of the action. Figure 3(a) illustrates the structure of the behavior cloning policy network. Since equilibrium strategies in most information sets are non-trivial probability distributions, we apply a softmax layer after the output layer to obtain the final mixed strategy.

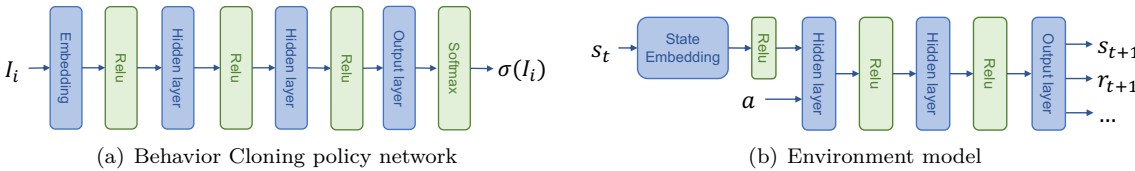

(a) Behavior Cloning policy network  (b) Environment model

Figure 3: Structure of neural networks

## 5.2 Model-Based Framework

Many model-based algorithms exist for offline single-agent RL; however, they cannot be directly applied to solve the OEF problem. The primary reason is their inherent reliance on the absence of strategic opponents in the environment. This means that if we use these algorithms in the OEF setting, we would need to train a model for each player to compute the best response strategy given any opponent strategy. This process would be extremely time-consuming and highly computationally demanding. To address this issue, we train a single environment model for all players instead of using the single-agent model-based algorithm for each player. The trained environment model can capture the necessary game information for evaluating any action tuple. In this manner, only one environment model needs to be trained, and all players can share this environment model to compute the equilibrium strategy.

### 5.2.1 Environment Model

In this section, we describe the methodology for training an environment model based on an OEF dataset. The trained environment model aims to provide all the information required for computing the equilibrium strategy for all players. As a result, the environment model can work as the game's environment, with the primary task of learning the game's dynamics. Considering that the dynamics of the game employed in this paper are relatively stable, we can implement supervised learning techniques to train the environment model. Figure 3(b) illustrates the structure of the environment model.

It should be noted that the offline dataset comprises data tuples $(s_t, a, s_{t+1}, r_{t+1})$, thereby enabling seamless training of the environment model using supervised learning methodologies based on the offline dataset. The environment model $E$, parameterized by $\theta_e$, takes the game state $s_t$ and action $a$ performed by the player at state $s_t$ as input, subsequently producing the next game state $s_{t+1}$ and rewards $r_{t+1}$ for all players. Depending on the specific scenario, additional game information can be predicted to facilitate the computation of equilibrium strategies, such as the legal action set for the subsequent state $A(s_{t+1})$ or the termination of the game. As delineated in Section 2.1, the chance player embodies stochastic events beyond the control of the player. Consequently, the game's dynamics are primarily driven by the chance player (player $c$). To handle this dynamism, the environment model also outputs whether the next state is played by the chance player. If so, an action is simply sampled according to the predicted legal action set. For training the environment model, stochastic gradient descent (SGD) is employed as the optimizer for parameter updates. Any loss function satisfying Bregman divergence conditions (Banerjee et al., 2005) can be utilized. In this paper, the mean squared error loss is employed and defined as follows,

$$\mathcal{L}_{env} = \mathbb{E}_{(s_t, a, s_{t+1}, r_{t+1}) \sim D}[MSE((s_{t+1}, r_{t+1}), E(s_t, a; \theta_e))].$$

Lastly, the environment model is trained by performing mini-batch SGD iterations.

### 5.2.2 Model-Based Algorithms

Once the environment model is adequately trained, it can provide sufficient game information for equilibrium computation. Utilizing the trained environment model, we propose a general model-based framework capable of generalizing any online equilibrium finding algorithm to the context of the OEF setting by substituting the actual environment with the trained environment model. To demonstrate the generalization of existing online equilibrium finding algorithms to the OEF setting, we instantiate three model-based algorithms: Offline Equilibrium Finding-Policy Space Response Oracles (OEF-PSRO), Offline Equilibrium Finding-Deep CFR

(OEF-CFR), and Offline Equilibrium Finding-Joint Policy Space Response Oracles (OEF-JPSRO). OEF-PSRO and OEF-CFR generalize PSRO and Deep CFR, respectively, to compute Nash Equilibria (NEs), while OEF-JPSRO generalizes JPSRO to compute Coarse Correlated Equilibria (CCEs).

In PSRO or JPSRO, a meta-game is represented as an empirical game that begins with a single policy (uniform random) and is iteratively expanded by adding new policies (oracles) approximating the best responses to the meta-strategies of other players. It is evident that when computing the best response policy oracle necessitates interaction with the environment to obtain game information. In the OEF setting, only an offline dataset is provided, rendering the direct application of PSRO or JPSRO unfeasible. In OEF-PSRO and OEF-JPSRO, the trained environment model substitutes the actual environment to supply the game information. It is widely acknowledged that when computing the best response policy using DQN or other RL algorithms, the next state and reward based on the current state and the action are required. The trained environment model can provide such information and additional details for approximating the missing entries in the meta-game matrix using the same approach. Deep CFR is a variant of CFR that employs neural networks to approximate counterfactual regret values and average strategies. This algorithm necessitates the partial traversal of the game tree to compute the counterfactual regret value, which in turn requires an environment to provide the necessary game information. Analogous to OEF-PSRO, OEF-CFR also utilizes the trained environment model to replace the actual environment. During the traversal, the environment must identify the next game state and utility for the terminal game state, for which the trained environment model is employed. These algorithms are elaborated in detail in Appendix E.

## 5.3 Combination Method: BCMB

Although the above two algorithms can be used to solve the OEF problem, they can only perform well in certain cases, as shown in the experiment section. To this end, we combine the behavior cloning technique and the model-based framework, creating a more robust approach for tackling the OEF problem.

We move to introduce the combination method **BCMB**, i.e., how to combine the two trained policies. Let $\alpha$ be the weight of the BC policy, making the weight of the MB policy $1 - \alpha$. The simplest way to select the parameter $\alpha$ is to randomly choose a number from 0 to 1. This method can be done in an offline way since it does not need any interaction with the actual environment. However, this method cannot guarantee to get the best parameter. In real life, if we can first interact with the actual environment, then we can use one online search method to select a better parameter as follows. We first preset

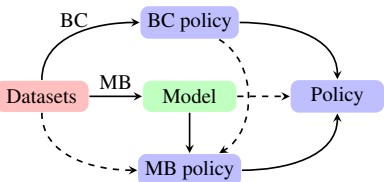

Figure 4: The flow of OEF algorithms.

11 weight assignment plans, i.e., $\alpha \in \{0, 0.1, 0.2, ..., 0.9, 1\}$. Next, we use these 11 weight assignment plans to combine these two policies, generating a set of final policies. Finally, we test these combination policies in the actual game to determine the best final policy based on the measure used to assess the gap from the equilibrium strategy. This method can get a good parameter while this method needs online interactions. To reduce online interactions, another method that sandwichs the above two methods is to train a parameter predictor model based on the difference between bc policy and mb policy. In this way, we first collect training data (the policy difference and the good parameter value) using the above online method for one game, then the parameter predictor can be trained based on these training data. Since the parameter predictor only depends on the difference between the two policies, it can also be used in different games (More details and experimental results can be found in Appendix E). Although this method can only provide an approximate best parameter, it needs little online interaction and can be reused in different games.

We present the general procedure of our OEF algorithm, BCMB, in Algorithm 1. Given the offline dataset $D$, we first train an environment model $E$ according to the method introduced in Section 5.2.1. Then, based on the trained environment model, we can obtain the MB policy using the model-based algorithm, which is the generalized algorithm from an online equilibrium finding algorithm under the model-based framework. To get the BC policy, we directly apply the behavior cloning technique on the offline dataset. Finally, we combine these two policies, i.e., the BC policy and the MB policy, by assigning appropriate weights to these two policies to derive the final policy. Figure 4 illustrates the whole structure of our OEF algorithm. These

---

**Algorithm 1** General Framework of Offline Equilibrium Finding algorithm

---

1: **Input:** an offline dataset $D$
2: Train an environment model $E$ based on the offline dataset $D$;
3: Learn a policy $\pi^{mb}$ on environment model $E$ using any model-based algorithm;
4: Learn a policy $\pi^{bc}$ based on the offline dataset $D$ using behavior cloning technique;
5: Combine $\pi^{bc}$ and $\pi^{mb}$ to get the policy $\pi$ by selecting the best $\alpha$ based on the test results in actual game;
6: **Output:** Policy $\pi$

---

dashed lines in the figure represent potential avenues for future research: i) whether we can learn an MB policy with the regularization of the BC policy, as well as interact with the dataset, and ii) if we can use the learned model to get the proper weights when combining these two policies.

### 5.4 Theoretical Analysis

To better understand our OEF problem and algorithm, we offer some theoretical analysis of these algorithms' performance under different offline datasets. To facilitate the analysis, we initially provide two assumptions regarding the data coverage of the random dataset and the expert dataset, respectively. These assumptions are derived from the generation process of these datasets and align with intuitive understanding. Since the random dataset is generated using the uniform strategy, the state-action pair would be covered as long as we generate enough data. Because the expert dataset is generated using the Nash equilibrium strategy, the strategy got from the expert dataset in a statistical way would be the Nash equilibrium strategy. Therefore, we can easily get the following assumption for these two datasets.

**Assumption 5.1.** *The **random dataset** satisfies the uniform dataset coverage assumption, i.e., for $\forall s_t$ and $\forall a \in A(s_t), (s_t, a, s_{t+1})$ is covered by the random dataset.*

**Assumption 5.2.** *The **expert dataset** only covers the state-action pair deduced by the Nash Equilibrium (NE) strategy and the frequency of these state-action pairs is corresponding to the NE strategy.*

Since the behavior cloning policy and the environment model are both neural network models, they are trained in a supervised learning manner. Therefore, we provide a general generalization bound for training such neural network models which can be found in Appendix D. Then, based on these sample analysis results, we provide some analysis results regarding the relationship between the above algorithms and the OEF datasets. Here, we only provide several key theoretical analysis results. Further results and proofs can be found in Appendix D. The behavior cloning technique possesses the capability to mimic the behavior policy of the dataset, with its performance primarily relying on the dataset's quality. Consequently, we present the following theorem to summarize the performance of BC under various datasets.

**Theorem 5.1.** *Assuming that the behavior cloning policy is trained on the offline dataset with an extremely small training error $\epsilon$, then the behavior cloning technique (BC) can get the equilibrium strategy under the expert dataset, and cannot get the equilibrium strategy under the random dataset.*

Since the trained environment model substitutes the actual environment in these algorithms, the performance primarily depends on the quality of the trained environment model. Consequently, we provide the following theorem to generalize the performance of the MB approach under varying datasets.

**Theorem 5.2.** *Assuming that the environment model is trained on the offline dataset with an extremely small training error $\epsilon$, then the model-based framework (MB) can converge to an equilibrium strategy under the random dataset and cannot guarantee to converge under the expert dataset.*

The BCMB algorithm combines the BC policy and the MB policy. As a result, drawing upon the insights from the two theorems above, we can readily derive the following theorem regarding its performance.

**Theorem 5.3.** *Under the assumptions in Theorems 5.1 and 5.2, BCMB can compute the equilibrium strategy under either the random dataset or the expert dataset.*

To better understand our OEF algorithm, we also provide a guarantee of the solution quality for our OEF algorithm under a more general case, as represented by the following theorem.

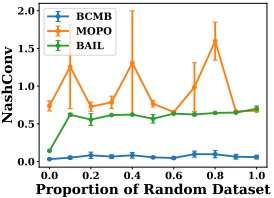 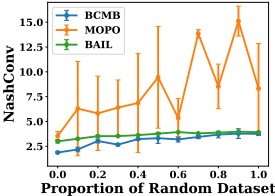 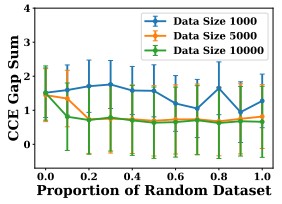 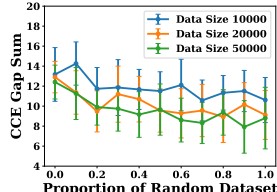

(a) Two-player Kuhn poker    (b) Two-player Leduc poker    (a) Three-player Kuhn Poker    (b) Three-player Leduc Poker

Figure 5: Comparison results with Offline RL     Figure 6: Experimental results on multi-player games.

**Theorem 5.4.** *Assuming that the environment model and the behavior cloning policy are trained with an extremely small training error $\epsilon$ on the offline dataset $\mathcal{D}_\sigma$ generated using $\sigma$, BCMB can get an equal or better strategy than $\sigma$.*

## 6 Experiments

To assess the performance of our OEF algorithms, we conduct the following experiments: i) we conduct two offline RL algorithms in the OEF setting to evaluate their performance; ii) we perform experiments on various offline datasets to evaluate the effectiveness of our algorithm in computing NEs under the OEF setting; and iii) we conduct experiments on two three-player games to assess the performance of our model-based algorithm in computing CCEs under the OEF setting.

### 6.1 Experimental Setting

OpenSpiel[1] is an extensive collection of environments and algorithms for research in games (Lanctot et al., 2019). We use it as our experimental platform, as it is widely accepted and implements many different games. In this paper, we select several poker games (Kuhn poker, Leduc poker), Liar's Dice, and Phantom Tic-Tac-Toe, which are all widely used in previous works (Lisý et al., 2015; Brown & Sandholm, 2019), as experimental domains. Firstly, we generate the OEF datasets for every game using the methods introduced in Section 4. Then we conduct our experiments on these OEF datasets. NashConv is used to measure how close the strategy is to NEs, and (C)CE Gap Sum is employed as a measurement of closeness to (C)CEs. All results are averaged over three seeds, and error bars are also reported. Only selected results are presented here. The remaining experiment results, ablation study, and parameter setting can be found in Appendix F.

### 6.2 Comparison with Offline RL

In this section, we provide empirical evidence demonstrating that naive offline RL algorithms are insufficient for solving the OEF problem. To support this claim, we choose one model-based offline RL algorithm – Model-based Offline Policy Optimization (MOPO) (Yu et al., 2020) and one model-free offline RL algorithm – Best-Action Imitation Learning (BAIL) (Chen et al., 2020) as the representative of offline RL algorithms. Figure 5 shows the comparison results between offline RL algorithms and our OEF algorithm in two-player Kuhn poker and two-player Leduc poker games. The x-axis represents the proportion of random data in the hybrid dataset. When the ratio is zero, the dataset is equivalent to the expert dataset; conversely, when the ratio is one, the hybrid dataset consists entirely of the random dataset. As shown in the figure, we observe that our algorithm outperforms the two offline RL algorithms. Additionally, we notice that the performance of the MOPO algorithm varies significantly across different datasets. Compared to the MOPO algorithm, the performance of the BAIL algorithm appears to be more closely related to the quality of the dataset. However, neither of these offline RL algorithms can produce a strategy profile close enough to the equilibrium strategy, which might be attributed to the players' policies being optimized independently.

---

[1]https://github.com/deepmind/open_spiel

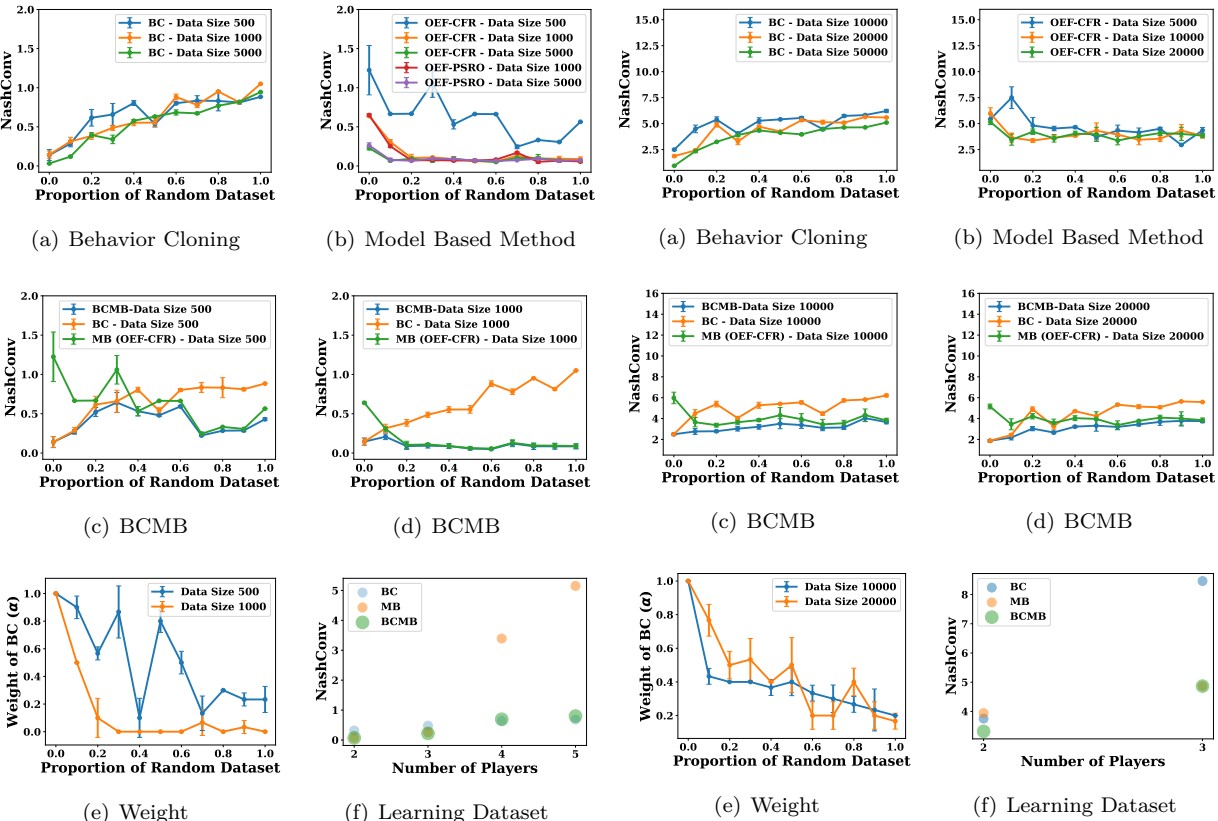

Figure 7: Experimental results on Kuhn poker.

Figure 8: Experimental results on Leduc poker.

## 6.3 Computation of Nash Equilibrium

We move to evaluate the performance of our OEF algorithm in computing the NE strategy. We first assess the individual performance of the behavior cloning technique and the model-based algorithm by applying them separately to several games. This assessment helps us understand the strengths and weaknesses of each algorithm. Figures 7(a) and 8(a) show the results of BC on two-player Kuhn poker and Leduc poker games. As the proportion of the random dataset increases, we observe that the performance of BC policy decreases in these two games. Additionally, we notice that as the size of offline data increases, the performance becomes more stable, while the improvement is not significant. This observation suggests that the performance of BC primarily depends on the quality of datasets, i.e., the quality of the behavior policy used to generate the dataset. Figures 7(b) and 8(b) depict the results of the MB framework. As shown in Figure 7(b), we observe that different model-based algorithms achieve nearly identical results. It indicates that the performance of the MB framework primarily relies on the quality of the trained environment model and is independent of the algorithm used to compute the equilibrium strategy. Another observation is that as the size of the offline dataset increases, the performance improves. It indicates that if the dataset includes sufficient data, the trained environment model is closer to the actual environment. Based on the above results, we can conclude that the BC performs poorly in the random dataset but well in the expert dataset. On the other hand, the MB framework exhibits slightly poorer performance in the expert dataset while performing well in the random dataset. This finding aligns with our theoretical analysis results, i.e., Theorem 5.1 and Theorem 5.2.

Finally, we proceed to evaluate the performance of our OEF algorithm – BCMB. Figures 7(c)-7(d) and 8(c)-8(d) present the results of our OEF algorithm on two-player Kuhn and Leduc poker games. For comparison, we also include the results of the BC and MB methods in these figures. We observe that our OEF algorithm outperforms both BC and MB methods in all cases, demonstrating the effectiveness of the combination. The optimal weights of BC policy ($\alpha$) for these combined policies, as illustrated in Figures 7(e) and 8(e), show

that as the proportion of the random dataset decreases, the weight of the BC policy in the combined policy increases. It also confirms that the BC policy performs better under the expert dataset and the MB policy performs better under the random dataset. We also evaluate our OEF algorithm on various poker games with different players using learning datasets, which can be considered as datasets generated by unknown strategies. Figures 7(f) and 8(f) demonstrate that our OEF algorithm outperforms other methods in all games. It indicates that given an OEF dataset generated by an unknown strategy, our OEF algorithm can consistently obtain a satisfactory approximate NE strategy.

### 6.4  Computation of Coarse Correlated Equilibrium

To evaluate the performance of our model-based framework in computing the CCE strategy, we apply the OEF-JPSRO algorithm to two three-player poker games using hybrid datasets. We do not perform the behavior cloning technique here since the offline dataset is collected using an independent strategy for each player, rather than a joint strategy. As described in Section 4, in multi-player games, although there is no guarantee of convergence to NE, we can still use PSRO with $\alpha$-rank as the meta-solver to get a fairly effective strategy for generating the expert dataset. Figure 6 represents the results for three-player Kuhn and Leduc poker games. We can observe that as the size of the offline data increases, the performance of OEF-JPSRO improves. This further supports the notion that the performance of the model-based framework primarily depends on the trained environment model and highlights its significance in solving the OEF problem.

## 7  Conclusion

We initiated an investigation into offline equilibrium finding (OEF), which focuses on finding equilibria in offline datasets. We first constructed OEF datasets from widely-used games using several data-collecting methods. To tackle the OEF problem, we proposed a model-based framework capable of generalizing any online equilibrium finding algorithm with minor changes by introducing an environment model. Specifically, we adapted several existing online equilibrium finding algorithms to the OEF setting to compute different equilibrium solutions. To further improve the performance, we combined the behavior cloning technique with the model-based framework. Experimental results demonstrated that our algorithm outperforms existing offline RL algorithms, and the model-based method is essential for the OEF setting. We hope our efforts will open new directions in equilibrium finding and accelerate the research in game theory.

**Future works.** There are several limitations of this work that we intend to tackle in the future. First, the games we considered are rather smaller and large-scale games like Texas Hold'em poker (Brown & Sandholm, 2018) were postponed till future work. Second, the types of generated offline datasets are limited. For future work, we plan to collect datasets using large-scale games and connect our library to StarCraft II Unplugged (Mathieu et al., 2021). We will also include more data-collecting strategies (e.g., bounded rational agents) as well as additional human expert data[2] to diversify the provided datasets.

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

## A  Frequently Asked Questions

**Q1: What is the impact of this work?**

Offline RL aims to bridge the gap between reinforcement learning and real-world applications. We anticipate that our offline equilibrium finding setting could inspire new research directions in equilibrium finding and pave a path to solving real-world problems using these game theory-based methods. Notably, offline RL algorithms cannot be directly applied to the OEF setting. Offline RL seeks to compute the optimal strategy from the single agent perspective, but this optimal strategy might be exploitable in a game setting. In such situations, the Nash Equilibrium (NE) strategy may be a more suitable solution, as it comprises non-exploitable strategies. Consequently, OEF plays a crucial role in obtaining more robust strategies for tackling these competitive real-world problems.

**Q2: How to connect the example scenario with offline equilibrium finding?**

In the example scenario, Player A aims to obtain a larger reward by employing the best strategy (i.e., the best response against Player B's previous policy). However, this best strategy may be exploited by Player B if he adapts his strategy accordingly. As a result, Player A must learn more about the game information by observing replays (e.g., actions and preferences of Player B). To minimize the risk of being exploited, the optimal solution for Player A is to choose the Nash equilibrium strategy of the underlying game.

**Q3: Why OEF is important and is more difficult than offline cooperative multi-agent RL?**

Utilizing OEF algorithms specifically designed for adversarial environments is crucial in strictly competitive games, such as security games. This setting fundamentally differs from offline multi-agent RL, which generally focuses on cooperation between agents rather than strict competition. For instance, consider the class of pursuit-evasion games, where the pursuer (defender) chases the evader (attacker). In this scenario, we cannot make any assumptions about the attacker's strategy beforehand, as the attacker is strategic and capable of learning. Employing a vanilla offline RL algorithm to learn the defender's optimal strategy based solely on historical data might lead to a significant utility loss, as the defender's optimal strategy could be exploitable. In other words, the attacker may switch to the best response against the computed strategy of the defender instead of adhering to their past behavior estimated from the data. Therefore, achieving Nash Equilibrium (NE) may be a more suitable solution, as NE strategies are non-exploitable.

To be more specific, traditional offline RL focuses on learning the optimal strategy, i.e., obtaining the highest utility, for an agent acting in a dynamic environment modeled as a single MDP, which does not depend on the actions of other agents. In contrast, in two-player games, the dynamics for one player depend not only on the environment but also on the strategy of the opponent. In other words, the MDP a player acts in games is determined by both the game and the fixed strategy of the opponent, and hence a change in the opponent's strategy instigates a corresponding change in the MDP. This makes computing the best strategy for the defender against a strategic opponent using offline RL significantly more difficult. The framework of OEF we introduced provides methods for computing a player's NE strategy, which is their optimal strategy against a strategic opponent (i.e., the worst case for the player).

**Q4: What are the differences between OEF and EGTA?**

1) As described in (Wellman, 2006), EGTA takes the game simulator as the fundamental input and performs strategic reasoning through interleaved simulation and game-theoretic analysis. Therefore, **the game simulator is required in EGTA.** In contrast, under the OEF setting, only the offline dataset is available and the game simulator is not required.

2) The estimated game model (empirical game) in EGTA is built based on the simulation's results, which are obtained by performing **known strategies** on the simulator. In contrast, in the OEF setting, the offline dataset is generated with an **unknown strategy**. In our work, although we use different behavior strategies to generate several offline datasets, we do not utilize these behavior strategies when performing our OEF algorithm.

Therefore, our proposed approach is different from EGTA. It is more challenging to find the equilibrium strategy in our OEF setting.

**Q5: What are the novelties of the proposed OEF algorithm – BCMB?**

We are the first ones to propose an empirical algorithm for solving the OEF problem. We introduce an environment model to propose a model-based framework that can generalize any existing online equilibrium finding algorithm to the context of the OEF setting. Due to the performance limitations of the model-based framework on certain offline datasets, we combine a model-free algorithm – the behavior cloning technique, with the model-based framework to improve performance. Unlike traditional offline RL algorithms, which belong to either model-based or model-free categories, our algorithm combines the advantages of both model-based and model-free approaches to efficiently solve the OEF problem.

## B   Related Work Overview

**Offline Reinforcement Learning (Offline RL).** Offline RL is a *data-driven* paradigm that learns exclusively from static datasets of previously collected interactions, making it feasible to extract policies from large and diverse training datasets (Levine et al., 2020). This paradigm can be extremely valuable in settings where online interaction is impractical, either because data collection is expensive or dangerous (e.g., in robotics (Singh et al., 2021), education (Singla et al., 2021), healthcare (Liu et al., 2020), and autonomous driving (Kiran et al., 2022)). Therefore, efficient offline RL algorithms have a much broader range of applications than online RL and are particularly appealing for real-world applications (Prudencio et al., 2022). Due to its attractive characteristics, there have been a lot of recent studies. Here, we can divide the research of Offline RL into two categories: model-based and model-free algorithms.

Model-free algorithms mainly use the offline dataset directly to learn a good policy. When learning the strategy from an offline dataset, we have two types of algorithms: actor-critic and imitation learning methods. Those actor-critic algorithms focus on implementing policy regularization and value regularization based on existing reinforcement learning algorithms. Haarnoja et al. (2018) propose soft actor-critic (SAC) by adding an entropy regularization term to the policy gradient objective. This work mainly focuses on policy regularization. For the research of value regularization, an offline RL method named Constrained Q-Learning (CQL) (Kumar et al., 2020) learns a lower bound of the true Q-function by adding value regularization terms to its objective. Another line of research on learning a policy is imitation learning which mimics the behavior policy based on the offline dataset. Chen et al. (2020) propose a method named Best-Action Imitation Learning (BAIL), which fits a value function, then uses it to select the best actions. Meanwhile, Siegel et al. (2020) propose a method that learns an Advantage-weighted Behavior Model (ABM) and uses it as a prior in performing Maximum a-posteriori Policy Optimization (MPO) (Abdolmaleki et al., 2018). It consists of multiple iterations of policy evaluation and prior learning until they finally perform a policy improvement step using their learned prior to extracting the best possible policy.

Model-based algorithms rely on the offline dataset to learn a dynamics model or a trajectory distribution used for planning. The trajectory distribution induced by models is used to determine the best set of actions to take at each given time step. Kidambi et al. (2020) propose a method named Model-based Offline Reinforcement Learning (MOReL), which measures their model's epistemic uncertainty through an ensemble of dynamics models. Meanwhile, Yu et al. (2020) propose another method named Model-based Offline Policy Optimization (MOPO), which uses the maximum prediction uncertainty from an ensemble of models. Concurrently, Matsushima et al. (2020) propose the BehaviorREgularized Model-ENsemble (BREMEN) method, which learns an ensemble of models of the behavior MDP, as opposed to a pessimistic MDP. In addition, it implicitly constrains the policy to be close to the behavior policy through trust-region policy updates. More recently, Yu et al. (2021a) proposed a method named Conservative Offline Model-Based policy Optimization (COMBO), a model-based version of CQL. The main advantage of COMBO concerning MOReL and MOPO is that it removes the need for uncertainty quantification in model-based offline RL approaches, which is challenging and often unreliable. However, these above Offline RL algorithms can not directly apply to the OEF problem, which we have described in Section 3 and experimental results empirically verify this claim.

**Empirical Game Theoretic Analysis (EGTA).** Empirical Game Theoretic Analysis is an empirical methodology that bridges the gap between game theory and simulation for practical strategic reasoning (Wellman, 2006). In EGTA, game models are iteratively extended through a process of generating new strategies based on learning from experience with prior strategies. The strategy exploration problem (Jordan et al., 2010) that how to efficiently assemble an efficient portfolio of policies for EGTA is the most challenging problem in EGTA.

Schvartzman & Wellman (2009b) deploy tabular RL as a best-response oracle in EGTA for strategy generation. They also build the general problem of strategy exploration in EGTA and investigate whether better options exist beyond best-responding to an equilibrium (Schvartzman & Wellman, 2009a). Investigation of strategy exploration was advanced significantly by the introduction of the Policy Space Response Oracle (PSRO) framework (Lanctot et al., 2017) which is a flexible framework for iterative EGTA, where at each iteration, new strategies are generated through reinforcement learning. Note that when employing NE as

the meta-strategy solver, PSRO reduces to the double oracle (DO) algorithm (McMahan et al., 2003). In the OEF setting, only an offline dataset is provided, and there is no accurate simulator. In EGTA, a space of strategies is examined through simulation, which means that it needs a simulator, and the policies are known in advance. Therefore, techniques in EGTA cannot directly apply to OEF.

**Opponent Modeling (OM) in Multi-Agent Learning.** Opponent modeling algorithm is necessary for multi-agent settings where secondary agents with competing goals also adapt their strategies, yet it remains challenging because policies interact with each other and change (He et al., 2016). One simple idea of opponent modeling is to build a model each time a new opponent or group of opponents is encountered (Zheng et al., 2018). However, it is infeasible to learn a model every time. A better approach is to represent an opponent's policy with an embedding vector. Grover et al. (2018) use a neural network as an encoder, taking the trajectory of one agent as input. Imitation learning and contrastive learning are also used to train the encoder. Then, the learned encoder can be combined with RL by feeding the generated representation into the policy or/and value network. DRON (He et al., 2016) and DPIQN (Hong et al., 2017) are two algorithms based on DQN, which use a secondary network that takes observations as input and predicts opponents' actions. However, if the opponents can also learn, these methods become unstable. So it is necessary to take the learning process of opponents into account.

Foerster et al. (2017) propose a method named Learning with Opponent-Learning Awareness (LOLA), in which each agent shapes the anticipated learning of the other agents in the environment. Further, the opponents may still be learning continuously during execution. Therefore, Al-Shedivat et al. (2017) propose a method based on a meta-policy gradient named Mata-MPG. It uses trajectories from current opponents to perform multiple meta-gradient steps and constructs a policy that favors updating the opponents. Meta-MAPG (Kim et al., 2021) extends this method by including an additional term that accounts for the impact of the agent's current policy on the future policies of opponents, similar to LOLA. Yu et al. (2021b) propose model-based opponent modeling (MBOM), which employs the environment model to adapt to various opponents. In the OEF setting, our goal is to compute the equilibrium strategy based on the offline dataset. Applying opponent modeling is not enough for calculating the equilibrium strategy in the OEF setting since the opponent will always best respond to the agent.

**Equilibrium Finding Algorithms.** The contemporary state-of-the-art algorithms for solving imperfect-information extensive-form games may be roughly divided into two groups: no-regret methods derived from CFR, and incremental strategy-space generation methods of the PSRO framework.

For the first group, CFR is a family of iterative algorithms for approximately solving large imperfect-information games. Let $\sigma_i^t$ be the strategy used by player $i$ in round $t$. We define $u_i(\sigma, h)$ as the expected utility of player $i$ given that the history $h$ is reached, and then all players act according to strategy $\sigma$ from that point on. Let us define $u_i(\sigma, h \cdot a)$ as the expected utility of player $i$ given that the history $h$ is reached and then all players play according to strategy $\sigma$ except player $i$ who selects action $a$ in history $h$. Formally, $u_i(\sigma, h) = \sum_{z \in Z} \pi^\sigma(h, z) u_i(z)$ and $u_i(\sigma, h \cdot a) = \sum_{z \in Z} \pi^\sigma(h \cdot a, z) u_i(z)$. The *counterfactual value* $v_i^\sigma(I)$ is the expected value of information set $I$ given that player $i$ attempts to reach it. This value is the weighted average of the value of each history in an information set. The weight is proportional to the contribution of all players other than $i$ to reach each history. Thus, $v_i^\sigma(I) = \sum_{h \in I} \pi_{-i}^\sigma(h) \sum_{z \in Z} \pi^\sigma(h, z) u_i(z)$. For any action $a \in A(I)$, the counterfactual value of action $a$ is $v_i^\sigma(I, a) = \sum_{h \in I} \pi_{-i}^\sigma(h) \sum_{z \in Z} \pi^\sigma(h \cdot a, z) u_i(z)$. The *instantaneous regret* for action $a$ in information set $I$ of iteration $t$ is $r^t(I, a) = v_{P(I)}^{\sigma^t}(I, a) - v_{P(I)}^{\sigma^t}(I)$. The *counterfactual regret* for action $a$ in $I$ of iteration $T$ is $R^T(I, a) = \sum_{t=1}^T r^t(I, a)$. In vanilla CFR, players use *Regret Matching* to pick a distribution over actions in an information set proportional to the positive cumulative regret of those actions. Formally, in iteration $T + 1$, player $i$ selects action $a \in A(I)$ according to probabilities

$$\sigma^{T+1}(I, a) = \begin{cases} \frac{R_+^T(I, a)}{\sum_{b \in A(I)} R_+^T(I, b)} & \text{if } \sum_{b \in A(I)} R_+^T(I, b) > 0, \\ \frac{1}{|A(I)|} & \text{otherwise,} \end{cases}$$

where $R_+^T(I, a) = \max\{R^T(I, a), 0\}$ because we are concerned about the cumulative regret when it is positive only. If a player acts according to regret matching in $I$ on every iteration, then in iteration $T$, $R^T(I) \leq \Delta_i \sqrt{|A_i|} \sqrt{T}$ where $\Delta_i = \max_z u_i(z) - \min_z u_i(z)$ is the range of utilities of player $i$. Moreover, $R_i^T \leq$

$\sum_{I \in \mathcal{I}_i} R^T(I) \le |\mathcal{I}_i| \Delta_i \sqrt{|A_i|} \sqrt{T}$. Therefore, $\lim_{T \to \infty} \frac{R_i^T}{T} = 0$. In two-player zero-sum games, if both players' average regret $\frac{R_i^T}{T} \le \epsilon$, their average strategies $(\overline{\sigma}_1^T, \overline{\sigma}_2^T)$ form a $2\epsilon$-equilibrium (Waugh et al., 2009). Some variants are proposed to solve large-scale imperfect-information extensive-form games. Some sampling-based CFR variants (Lanctot et al., 2009; Gibson et al., 2012; Schmid et al., 2019) are proposed to effectively solve large-scale games by traversing a subset of the game tree instead of the whole game tree. With the development of deep learning techniques, neural network function approximation is also applied to the CFR algorithm. Deep CFR (Brown et al., 2019), Single Deep CFR (Steinberger, 2019), and Double Neural CFR (Li et al., 2019) are algorithms using deep neural networks to replace the tabular representation in the CFR algorithm.

For the second group, PSRO (Lanctot et al., 2017) is a general framework that scales Double Oracle (DO) (McMahan et al., 2003) to large extensive-form games via using reinforcement learning to compute the best response strategy approximately. To make PSRO more effective in solving large-scale games, Pipeline PSRO (P2SRO) (McAleer et al., 2020) is proposed by parallelizing PSRO with convergence guarantees. Extensive-Form Double Oracle (XDO) (McAleer et al., 2021) is a version of PSRO where the restricted game allows mixing population strategies not only at the root of the game but every information set. It can guarantee to converge to an approximate NE in a number of iterations that are linear in the number of information sets, while PSRO may require a number of iterations exponential in the number of information sets. Neural XDO (NXDO) as a neural version of XDO learns approximate best response strategies through any deep reinforcement learning algorithm. Recently, Anytime Double Oracle (ADO) (McAleer et al., 2022), a tabular double oracle algorithm for 2-player zero-sum games is proposed to converge to a Nash equilibrium while decreasing exploitability from one iteration to the next. Anytime PSRO (APSRO) as a version of ADO calculates best responses via reinforcement learning algorithms. Except for NEs, other equilibrium solution concepts, for example, (Coarse) Correlated equilibrium ((C)CE) is considered. Joint Policy Space Response Oracles (JPSRO) (Marris et al., 2021) is proposed for training agents in n-player, general-sum extensive-form games, which provably converges to (C)CEs. The excellent performance of these equilibrium finding algorithms depends on the existence of efficient and accurate simulators. However, constructing a sufficiently accurate simulator may not be feasible or very expensive. In this case, we may resort to offline equilibrium finding (OEF) where the equilibrium strategy is computed based on the previous game data.

## C   Visualization of Datasets

The additional figures provided showcase more visualization results for different datasets across various games. These results are consistent with those presented in the main paper. There are more high-frequency data in the expert dataset and the distributions of these datasets are very different.

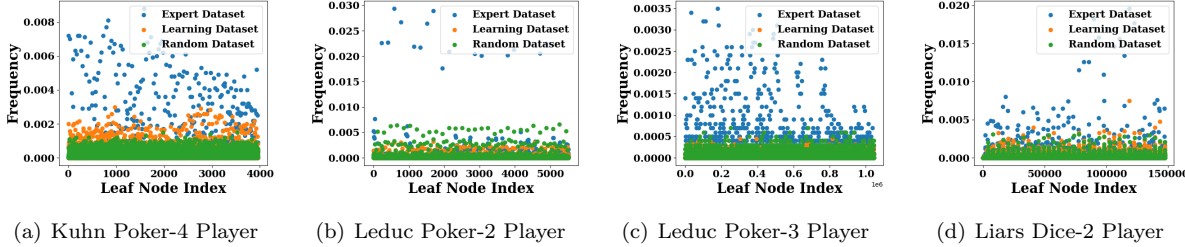

(a) Kuhn Poker-4 Player    (b) Leduc Poker-2 Player    (c) Leduc Poker-3 Player    (d) Liars Dice-2 Player

Figure 9: Frequency of leaf node for different offline datasets

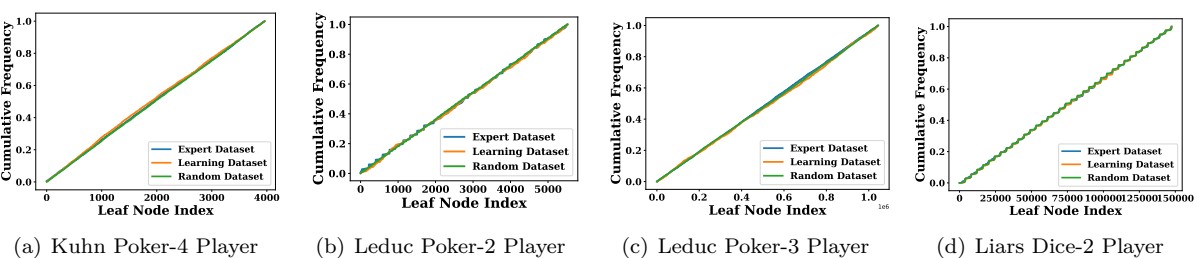

(a) Kuhn Poker-4 Player    (b) Leduc Poker-2 Player    (c) Leduc Poker-3 Player    (d) Liars Dice-2 Player

Figure 10: Cumulative frequency of leaf node for different offline datasets

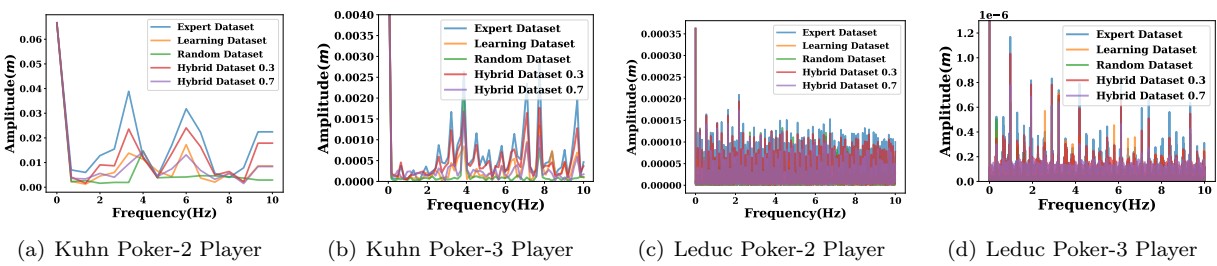

(a) Kuhn Poker-2 Player    (b) Kuhn Poker-3 Player    (c) Leduc Poker-2 Player    (d) Leduc Poker-3 Player

Figure 11: Amplitude-Frequency curve for different offline datasets

# D Theoretical Analysis

The concurrent works Cui & Du (2022); Zhong et al. (2022) investigate the necessary properties of offline datasets of two-player zero-sum Markov games to successfully infer their NEs. To do this, they proposed several dataset coverage assumptions. Following their assumptions Cui & Du (2022), we also define some hypotheses on the dataset coverage under our OEF setting and provide extensive analysis about how the dataset coverage influences computing the equilibrium under the OEF setting. Our results are mainly for computing Nash equilibrium in extensive-form games. Here, we did not provide a sample complexity analysis since the influence of dataset coverage on the algorithm is more important for our problem. The analysis of the dataset coverage can provide more intuitive insight into our algorithm.

## D.1 Minimal Dataset Assumption for the OEF problem

We first introduce the difference between the OEF and the Offline RL from the standpoint of a theoretical analysis of dataset coverage. As demonstrated in offline RL papers (Rashidinejad et al., 2021; Xie et al., 2021), a coverage condition over the optimal policy is sufficient for the offline learning of MDPs. Therefore, it is straightforward to extend this coverage condition to our OEF settings. The following assumption shows this extended coverage condition.

**Assumption D.1.** *(Single Strategy Coverage) The Nash equilibrium strategy $\sigma^*$ is covered by the dataset.*

Subsequently, a question arises: is the single strategy coverage assumption over the offline dataset also sufficient for computing NE strategy under the OEF setting? The answer is no, and we employ the following theorem to elucidate the rationale behind this.

**Theorem D.1.** *Single strategy coverage assumption over offline dataset is not sufficient for computing an NE strategy.*

*Proof.* We provide a counter-example to prove this theorem. Here, we consider two two-player extensive-form games $M_1$ and $M_2$, which are represented in Figure 12.

We can easily find that the NE of the game $M_1$ is strategy profile $\sigma^1 = (\sigma_1^1, \sigma_2^1) = (\{S_1 : a_1\}, \{S_2 : b_1\})$, i.e., player 1 plays $a_1$ at information set $S_1$ and player 2 plays $b_1$ at information set $S_2$. The NE of the game $M_2$ is strategy profile $\sigma^2 = (\sigma_1^2, \sigma_2^2) = (\{S_1 : a_2\}, \{S_2 : b_2\})$. Now we consider an offline dataset $D$ which is generated using a strategy profile $\sigma_D$ and the $\sigma_D$ is set to be the uniform distribution on the strategy profiles $\sigma^1$ and $\sigma^2$.

The dataset $D$ covers strategy profile $\sigma^1$ and $\sigma^2$. Therefore, the dataset $D$ satisfies the single strategy coverage assumption for these two games $M_1$ and $M_2$. However, it is impossible for any algorithm to distinguish these two extensive-form games only based on the dataset $D$ since these two games are both consistent on the dataset $D$.

Therefore, the single strategy converges assumption over the offline dataset is not sufficient for computing an NE strategy. □

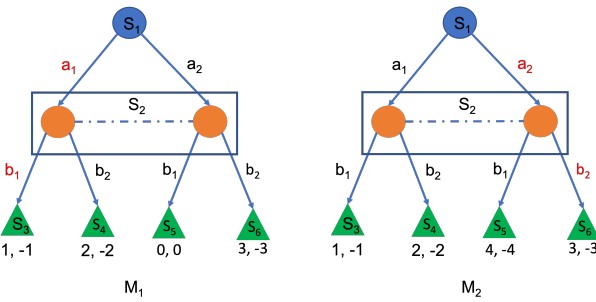

Figure 12: Example of two-player extensive-form game

From the above proof, we know that the single strategy coverage assumption over the dataset is sufficient for computing the optimal strategy under the offline RL setting while it is not sufficient for computing an NE strategy under the OEF setting. The intuition behind this theorem is that in an offline RL setting, we can easily use the data of two actions to decide which action is better, whereas, in an OEF setting, we cannot use data from only two action pairs to know which action pair is closer to NE, because identifying NE requires other action pairs as inferences. Based on this analysis, Cui & Du (2022) et al. provide a minimal coverage assumption over the dataset which is sufficient for computing an NE strategy in the two-player zero-sum Markov games, which is defined as follows,

**Assumption D.2.** *(Unilateral Coverage) For all strategy $\sigma_i$, $(\sigma_i, \sigma^*_{-i})$ for all player $i$ are covered by the dataset, where $\sigma^* = (\sigma^*_1, ..., \sigma^*_n)$ is the NE strategy.*

**Assumption D.3.** *(Deterministic Unilateral Coverage) For all deterministic strategy $\sigma_i$, $(\sigma_i, \sigma^*_{-i})$ for all player $i$ are covered by the dataset, where $\sigma^* = (\sigma^*_1, ..., \sigma^*_n)$ is the NE strategy.*

We can find that deterministic unilateral coverage assumption is equivalent to unilateral coverage assumption. The intuition behind this is that any mixed strategy can be represented by a combination of several deterministic strategies. Therefore, if all deterministic strategies are covered by the dataset, then all mixed strategies are also covered. Based on this, in the following proof, we only consider all deterministic strategies.

Cui & Du (2022) have proved that unilateral coverage assumption is the minimal assumption that is sufficient for computing an NE strategy in the two-player zero-sum Markov games. However, this conclusion is not hold for our model-based framework in computing the equilibrium strategy under the OEF setting. In other words, under the OEF setting, our model-based algorithm cannot guarantee coverage to the equilibrium strategy of the underlying game based on the dataset satisfying the unilateral coverage assumption.

**Theorem D.2.** *The unilateral coverage assumption over the offline dataset is not sufficient for our model-based algorithm to converge to the equilibrium strategy of the underlying game in the OEF setting.*

*Proof.* We prove it by providing a counter-example. Here, we consider an imperfect-information extensive-form game $M_3$, which is represented in Figure 13. We can easily find that the NE strategy of game $M_3$ is the strategy profile $\sigma^* = (\sigma_1, \sigma_2) = (\{S_1 : a_1\}, \{S_2 : b_1\})$.

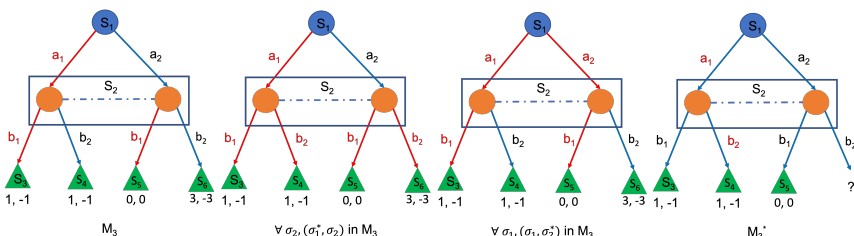

Figure 13: Example of two-player extensive-form game

To build an offline dataset satisfying the unilateral coverage assumption, the dataset needs to cover $(\sigma^*_1, \sigma_2)$ for all $\sigma_2$ and $(\sigma_1, \sigma^*_2)$ for all $\sigma_1$. We show the state-action pairs covered by these strategy profiles in Figure 13. These red lines show these covered state-action pairs. It means that the dataset satisfying the unilateral coverage assumption would cover these state-action pairs. When applying our model-based framework, the first step is to train an environment model based on the offline dataset. Assume that the environment model can be trained well which means that the environment model can precisely represent all these state-action pairs in the dataset. Therefore, the game represented by the trained environment model would be $M_3^*$ in Figure 13. Note that there are missing data in the game. Although our trained environment model can give approximate results for these missing data, it may result in a different equilibrium strategy. For example, if the missing value in $M_3^*$ is $(0,0)$ or $(-1,1)$, then the strategy profile $\sigma = (\sigma_1, \sigma_2^{'}) = (\{S_1 : a_1\}, \{S_2 : b_2\})$ would be the NE strategy of game $M_3^*$. However, the strategy profile $\sigma$ is not the NE strategy for the original game $M_3$. Therefore, the unilateral coverage assumption over the offline dataset is not sufficient for our model-based framework to converge to the NE strategy of the underlying game. $\square$

Therefore, the unilateral coverage assumption is not sufficient for our model-based framework to converge to the equilibrium strategy. To guarantee the convergence of our model-based framework, we provide a minimal dataset coverage assumption for our model-based algorithm to converge to the equilibrium strategy of the underlying game under the OEF setting.

**Assumption D.4.** *(Uniform Coverage) For all state $s_t$ and all actions $a_t \in A(s_t)$, all state-action pairs $(s_t, a_t, s_{t+1})$ are covered by the dataset.*

**Theorem D.3.** *The uniform coverage assumption over the offline dataset is the minimal dataset coverage assumption which is sufficient for our model-based algorithm to converge to the equilibrium strategy in the OEF setting.*

*Proof.* From the example in the proof of Theorem D.2, we find that a slight violation of the uniform coverage assumption will impede the computation of the NE strategy using our model-based algorithm. In other words, any state-action pair that is not covered by the dataset would impede the restructure of the game using our environment model.

Once the dataset satisfies the uniform coverage, then it covers all the state-action pairs in the game which is enough for training the environment model. It means that the environment model would be the same as the underlying game of the dataset. Then applying our model-based equilibrium finding algorithm on the trained environment model definitely can converge to the equilibrium strategy of the underlying game in the OEF setting. □

Here, we proved that the uniform dataset coverage assumption is sufficient for our model-based framework to converge to the equilibrium strategy. From the proof of Theorem D.2, we find that the game represented by the dataset satisfying the unilateral coverage assumption may be a part of the original game (here, we call the game in the dataset subgame). However, the non-uniqueness of the equilibrium in the subgame would result in the failure to find the equilibrium strategy of the underlying game using our model-based framework. The following theorem provides a more general analysis of the unilateral coverage assumption in the OEF setting.

**Theorem D.4.** *Under the assumption that the equilibrium strategy profile of the game represented by the dataset is unique, the unilateral coverage assumption would be the minimal assumption over the offline dataset which is sufficient for computing an NE strategy in the OEF setting.*

*Proof.* Firstly, we prove that a slight violation of the unilateral coverage assumption will impede the computation of the Nash equilibrium strategy. We can reuse the example game $M_1$ in the proof of Theorem D.1 and consider another dataset $D$ which is generated using strategy profile $\sigma_D$ and $\sigma_D$ is set to be the uniform distribution on these three deterministic strategy profiles $\sigma^1 = (\sigma_1^1, \sigma_2^1) = (\{S_1 : a_1\}, \{S_2 : b_1\})$, $\sigma^2 = (\sigma_1^2, \sigma_2^2) = (\{S_1 : a_2\}, \{S_2 : b_1\})$ and $\sigma^3 = (\sigma_1^1, \sigma_2^2) = (\{S_1 : a_2\}, \{S_2 : b_2\})$. Since the NE strategy of game $M_1$ is strategy profile $\sigma^1 = (\sigma_1^1, \sigma_2^1) = (\{S_1 : a_1\}, \{S_2 : b_1\})$, we can find that only the deterministic strategy profile $\sigma^4 = (\sigma_1^2, \sigma_2^1) = (\{S_1 : a_2\}, \{S_2 : b_1\})$ is not covered by the dataset $D$ compared with the dataset satisfying the unilateral coverage assumption. Then the game generated by the dataset is represented in Figure 14. we can find that the game generated based on the dataset has the unique

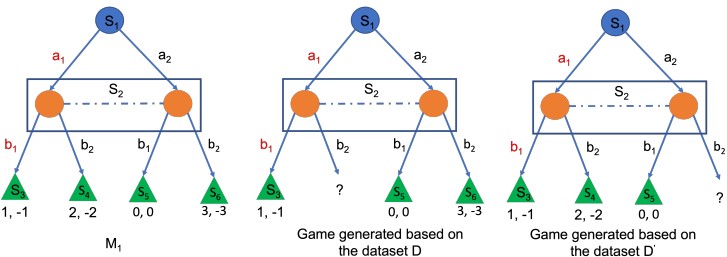

Figure 14: Example game

equilibrium strategy $\sigma^1 = (\sigma_1^1, \sigma_2^1) = (\{S_1 : a_1\}, \{S_2 : b_1\})$. Therefore, the dataset $D$ satisfies the assumption the game generated based on the dataset has a unique equilibrium and slightly violates the unilateral coverage assumption. However, we find that the different missing data values in the game generated based on the dataset would result in a different equilibrium strategy. For example, if the missing value in the game generated based on the dataset is $(0, 0)$, then the equilibrium strategy profile of the game would be $\sigma^* = (\sigma_1, \sigma_2) = \{S_1 : \{a_1 : 0.75, a_2 : 0.25\}, S_2 : \{b_1 : 0.75, b_2 : 0.25\}\}$, which is not the equilibrium strategy of the original game. Therefore, a slight violation of the unilateral coverage assumption will impede the computation of the equilibrium strategy.

Then we prove that the unilateral coverage assumption is sufficient for computing an NE strategy in the OEF setting under the unique equilibrium assumption. Recall the definition of NE strategy, the strategy profile $\sigma^*$ forms an NE strategy if $u_i(\sigma^*) \geq u_i(\sigma_i', \sigma_{-i}^*), \forall i \in N, \forall \sigma_i' \in \Sigma_i$ which means that $\sigma_i^*$ is the best response strategy against $\sigma_{-i}^*$ for $\forall i \in N$. According to the unilateral coverage assumption, the dataset covers all strategy profiles $(\sigma_i, \sigma_{-i}^*)$ for all $i$ and all $\sigma_i$. Then it is easy to verify which strategy for player $i$ is the best response strategy against $\sigma_{-i}^*$ based on the dataset. In other words, we have enough information about $(\sigma_i, \sigma_{-i}^*), \forall \sigma_i$ which is sufficient to verify that $\sigma_i^*$ is the best response strategy of $\sigma_{-i}^*$. In this way, we can verify the best response strategy for every player. Due to the uniqueness of the equilibrium strategy, the strategy $\sigma^*$ would also be the equilibrium strategy of the original game. We can give an example to further explain it. Consider another dataset $D'$ for the game $M_1$ which is generated using the strategy profile $\sigma_D'$ and $\sigma_D'$ is set to be the uniform distribution on these three deterministic strategy profiles $\sigma^1 = (\sigma_1^1, \sigma_2^1) = (\{S_1 : a_1\}, \{S_2 : b_1\})$, $\sigma^2 = (\sigma_1^2, \sigma_2^2) = (\{S_1 : a_2\}, \{S_2 : b_1\})$ and $\sigma^3 = (\sigma_1^1, \sigma_2^2) = (\{S_1 : a_1\}, \{S_2 : b_2\})$. We can easily verify that the dataset satisfies the unilateral coverage assumption for the game $M_1$ and the game generated based on the dataset $D'$ (Figure 14) has a unique equilibrium strategy, $\sigma^1 = (\sigma_1^1, \sigma_2^1) = (\{S_1 : a_1\}, \{S_2 : b_1\})$. Then we can find that whatever the missing value in the game is the equilibrium of the game would not change and is the same as the equilibrium strategy of the original game. Therefore, based on the above analysis, under the strong assumption (equilibrium uniqueness), the unilateral coverage assumption would be the minimal dataset coverage assumption. □

The above theorem proves that under the strong assumption (equilibrium uniqueness), the dataset satisfying the unilateral coverage assumption is sufficient for the computation of equilibrium strategy under the OEF setting. However, in the general OEF setting, to guarantee convergence under the dataset satisfying unilateral coverage assumption, it may need a more powerful algorithm that can solve the non-uniqueness of the equilibrium problem. We left it as future work.

### D.2 Generalization Bound for training Neural Network Model

In this paper, we need to train the behavior cloning policy model and environment model which are both neural network models. Both models are trained in a supervised learning manner with different loss functions. Here, we provide a general generalization bound for training such neural network models.

The supervised learning framework includes a data-generation distribution $\mathcal{D}$ a hypothesis class $\mathcal{H}$ of the neural network approximator, a training dataset $S$, and evaluation metrics to evaluate the performance of any approximator. Here, we use the loss function $l$ to evaluate the performance of any approximation. The learning framework aims to minimize the true risk function $L_{\mathcal{D}}(h)$ which is the expected loss function of $h$ under the distribution $\mathcal{D}$.

$$L_{\mathcal{D}}(h) = \mathbb{E}_{d \sim \mathcal{D}}[l(h(d), d)]$$

Accordingly, the empirical risk function $L_S(h)$ on the training dataset $S$ can be defined as:

$$L_S(h) = \frac{1}{|S|} \sum_{d \sim S} [l(h(d), d)] \tag{1}$$

To get a generalization bound, we use an auxiliary lemma from Shalev-Shwartz & Ben-David (2014). Therefore, we can measure the capacity of the composition function class $l \circ \mathcal{H}$ using the empirical Rademacher

complexity on the training set $S$ with size $m$, which is defined as:

$$\mathcal{R}_S(l \circ \mathcal{H}) = \frac{1}{m} \mathbb{E}_{\mathbf{x} \sim \{+1,-1\}^m} [\sup_{\sigma \in \Sigma} \sum_{i=1}^{m} x_i \cdot l(h(d_i), d_i)]$$

where $\mathbf{x}$ is distributed i.i.d. according to uniform distribution in $\{+1, -1\}$. Then we have the following lemma from Shalev-Shwartz & Ben-David (2014).

Before providing the generalization bound, we first provide the distance between two different approximators and one common theorem to facilitate the proof of the generalization bound.

**Definition D.1.** ($r$-cover). We say function class $\mathcal{H}_r$ $r-$cover $\mathcal{H}$ under $\ell_{\infty,1}$-distance if for all function $h \in \mathcal{H}$, there exists $h_r$ in $\mathcal{H}_r$ such that $||h - h_r||_{\infty,1} = \max_{x \in \mathcal{D}} ||h(x) - h_r(x)||_1 \leq r$.

**Definition D.2.** ($r$-covering number). The $r$-covering number of $\mathcal{H}$, denoted by $\mathcal{N}_{\infty,1}(\mathcal{H}, r)$, is the cardinality of the smallest function class $H_r$ that $r$-covers $\mathcal{H}$ under $\ell_{\infty,1}$-distance.

**Theorem D.5.** *(Shalev-Shwartz & Ben-David (2014)) Let $S$ be a training set of size $m$ drawn i.i.d. from distribution $\mathcal{D}$. Then with probability of at least $1 - \delta$ over draw of $S$ from $\mathcal{D}$, for all $\sigma \in \Sigma$,*

$$L_{\mathcal{D}}(h) - L_S(h) \leq 2\mathcal{R}_S(l_{bc} \circ \Sigma) + 4\sqrt{\frac{2 \ln (4/\delta)}{m}}$$

Here, we provide the generalization bound to measure the generalizability of the trained approximator under a training dataset with size $m$.

**Theorem D.6** (Generalization bound). *Assume that the loss function $l$ is $T$-Lipschitz continuous, then for hypothesis class $\mathcal{H}$ of approximator and distribution $\mathcal{D}$, with probability at least $1-\delta$ over draw of the training set $S$ with size $m$ from $\mathcal{D}$, $\forall h \in \mathcal{H}$ we have*

$$L_{\mathcal{D}}(h) - L_S(h) \leq 2 \cdot \inf_{r>0} \left[ \frac{\sqrt{2 \log \mathcal{N}_{\infty,1}(\mathcal{H}, r)}}{m} + Tr \right] + 4\sqrt{\frac{2 \ln (4/\delta)}{m}}.$$

*Proof.* According to Theorem D.5, we have

$$L_{\mathcal{D}}(h) - L_S(h) \leq 2\mathcal{R}_S(l \circ \mathcal{H}) + 4\sqrt{\frac{2 \ln (4/\delta)}{m}}.$$

We assume the loss function $l(x, y)$ is $T$-Lipschitz continuous under $\ell_k$-distance. Therefore,

$$|l(x, y) - l(x', y)| \leq T||x - x'||_k,$$

where $||\cdot||_k$ is the $k$-norm. Let $\mathcal{H}_r$ be the function class that $r$-cover $\mathcal{H}$ for some $r > 0$ and $|\mathcal{H}_r| = \mathcal{N}_{\infty,1}(\mathcal{H}, r)$ be the cardinality of the smallest function class $\mathcal{H}_r$. $\forall h \in \mathcal{H}$, denote $h_r \in \mathcal{H}_r$ be the function approximator that $r$-covers $h$. Based on above equation, we have

$$|l(h(x), y) - l(h_r(x), y)| \leq T||h(x) - h_r(x)||_k \leq Tr.$$

Then we have

$$\mathcal{R}_S(l \circ \mathcal{H}) \tag{2}$$

$$= \frac{1}{m}\mathbb{E}_{\mathbf{x}\sim\{+1,-1\}^m}\big[\sup_{h\in\mathcal{H}}\sum_{i=1}^{m}x_i \cdot l(h(d_i),d_i)\big] \tag{3}$$

$$= \frac{1}{m}\mathbb{E}_{\mathbf{x}\sim\{+1,-1\}^m}\big[\sup_{h\in\mathcal{H}}\sum_{i=1}^{m}x_i \cdot (l(h_r(d_i),d_i) + l(h(d_i),d_i) - l(h_r(d_i),d_i))\big] \tag{4}$$

$$\leq \frac{1}{m}\mathbb{E}_{\mathbf{x}\sim\{+1,-1\}^m}\big[\sup_{h_r\in\mathcal{H}_r}\sum_{i=1}^{m}x_i \cdot l(h_r(d_i),d_i)\big] + \frac{1}{m}\mathbb{E}_{\mathbf{x}\sim\{+1,-1\}^m}\big[\sup_{h\in\mathcal{H}}\sum_{i=1}^{m}|x_i \cdot Tr|\big] \tag{5}$$

$$\leq \sup_{h_r\in\mathcal{H}_r}\sqrt{\sum_{i=1}^{m}(\ell(h_r,d_i))^2} \cdot \frac{\sqrt{2\log\mathcal{N}_{\infty,1}(\mathcal{H},r)}}{m} + \frac{Tr}{m}\mathbb{E}_{\mathbf{x}}||\mathbf{x}||_1 \tag{6}$$

$$\leq \frac{\sqrt{2\log\mathcal{N}_{\infty,1}(\mathcal{H},r)}}{m} + Tr \tag{7}$$

The reduction from Eq. 19(a) to Eq. 19(j) is based on Massart's lemma Shalev-Shwartz & Ben-David (2014). From above, we can get

$$L_{\mathcal{D}}(h) - L_S(h) \leq 2\mathcal{R}_S(l \circ \mathcal{H}) + 4\sqrt{\frac{2\ln(4/\delta)}{m}} \tag{8}$$

$$\leq 2 \cdot \inf_{r>0}\big[\frac{\sqrt{2\log\mathcal{N}_{\infty,1}(\mathcal{H},r)}}{m} + Tr\big] + 4\sqrt{\frac{2\ln(4/\delta)}{m}} \tag{9}$$

$$\square$$

From the above theorem, we can find that given a training dataset with size $m$, we can have a generalization bound for the error depending on the characteristic of the loss function. In this paper, we follow the above supervised-learning framework to train the behavior cloning policy and environment model. Therefore, we can provide some assumptions for the trained policy and environment models based on the above theorem, as follows:

**Assumption D.5.** *If the error for training behavior cloning policy is less than an extremely small $\epsilon$ on the dataset with enough data (the size of data can be computed according to the above theorem), then we can consider that the trained behavior cloning policy model is the same as the underlying strategy of the dataset.*

**Assumption D.6.** *If the error for training the environment model is less than an extremely small $\epsilon$ on the dataset with enough data, then we can consider that the trained environment model is the same as the underlying game of the dataset.*

### D.3 Theoretical Guarantee for Our OEF Algorithm

We move to analyze our proposed datasets and their influences on our OEF algorithm. Here, we first provide two assumptions on our proposed datasets based on the generation process of the dataset.

Since we only use the NE strategy to generate the expert dataset, we can have the following assumption.

**Assumption D.7** (Assumption 5.2)**.** *The expert dataset only covers the state-action pair deduced by the Nash Equilibrium (NE) strategy and the frequency of these state-action pairs is corresponding to the NE strategy.*

Note that according to the above assumption, we can find that although the expert dataset satisfies the single strategy coverage assumption, it is more strict than the single strategy coverage assumption since the expert dataset only covers the NE strategy. From the empirical results on the expert dataset, we found that the model-based algorithm indeed cannot converge to the NE strategy. However, the behavior cloning

algorithm can get a good strategy on the expert dataset since it can mimic the strategy used to generate the expert dataset, i.e., the NE strategy.

The random dataset is sampled by the uniform strategy. Therefore, it would involve all the state transitions and we can have the following assumption for the random dataset.

**Assumption D.8** (Assumption 5.1)**.** *The random dataset satisfies the uniform dataset coverage assumption, i.e., for $\forall s_t$ and $\forall a \in A(s_t)$, $(s_t, a, s_{t+1})$ is covered by the random dataset.*

Since the random dataset satisfies the uniform dataset coverage assumption, according to Theorem D.3, the random dataset is sufficient for our model-based algorithm to compute the NE strategy. From the empirical results, we can find that the model-based algorithm performs best under the random dataset, which verifies that the random dataset is sufficient for computing the NE strategy. Next, we will provide more analysis of the relationship between the algorithm and the dataset.

From the empirical analysis, we find that the performance of the model-based algorithm mainly depends on the gap between the trained environment model and the actual game environment. It means that if the trained environment model can recover all the dynamics of the actual game, then the performance is good. Otherwise, the performance is worse. Since our model-based framework can generalize existing equilibrium finding algorithms to the context of the OEF setting and the performance of the existing equilibrium finding algorithm would also determine the convergence of the equilibrium strategy, we assume that there always exists an equilibrium finding algorithm for any game which can converge to the equilibrium strategy in the following proof. Then we have the following theorem.

**Theorem D.7** (Theorem 5.2)**.** *Assuming that the environment model is trained on the offline dataset with an extremely small training error $\epsilon$, then the model-based framework (MB) can converge to an equilibrium strategy under the random dataset and cannot guarantee to converge under the expert dataset.*

*Proof.* Since the training error of the environment model is less than $\epsilon$, the trained environment model can fully represent the information of the offline dataset according to Assumption D.6. If the random dataset is the offline dataset, the game defined by the trained environment model is the same as the actual game. The reason is that every state transition is covered by the random dataset according to Assumption D.8. Then the strategy learned by our model-based equilibrium finding algorithm is the approximate equilibrium strategy of the actual game due to the convergence property of the original equilibrium finding algorithm. Therefore, the model-based framework can converge to an equilibrium strategy under the random dataset.

If the offline dataset is the expert dataset, then the dataset only covers these state transitions related to the NE strategy according to Assumption D.7. Therefore, the state transition of the actual game may not be covered by the expert dataset. The environment model trained based on the expert dataset would produce different transition information on these states not shown in the dataset compared with the actual game. It would cause a gap between the trained environment model and the actual game. Although the model-based framework can learn an approximate equilibrium strategy of the game defined by the environment model, there is no guarantee that the learned strategy is the equilibrium strategy of the actual game. □

Theorem D.7 is consistent with our previous conclusion that single strategy coverage is insufficient for NE identification and dataset coverage satisfying Assumption D.4 is sufficient for NE identification according to Theorem D.3. And our empirical results also verify these conclusions. The model-based framework performs best under the random dataset and worst under the expert dataset.

Although the expert dataset satisfies the single strategy coverage, the expert dataset assumption is more strict than the single strategy coverage. We find that the behavior cloning algorithm can perform well on the expert dataset. Therefore, to offset the drawback of the model-based algorithm under the expert dataset, we propose to combine the behavior cloning (BC) technique. From the introduction of the BC technique, we know that the BC can mimic the behavior policy in the dataset. Therefore, we have the following theorem describing the power of the BC technique.

**Theorem D.8** (Theorem 5.1)**.** *Assuming that the behavior cloning policy is trained on the offline dataset with an extremely small training error $\epsilon$, then the behavior cloning technique (BC) can get the equilibrium strategy under the expert dataset, and cannot get the equilibrium strategy under the random dataset.*

*Proof.* The assumption that the behavior cloning policy is trained on the offline dataset with an extremely small training error $\epsilon$ means that the behavior cloning policy can precisely mimic the behavior strategy used to generate the offline dataset according to Assumption D.5. If the offline dataset is the expert dataset, according to Assumption D.7, the behavior strategy used to generate the expert dataset is the NE strategy. Therefore, applying the behavior cloning algorithm on the expert dataset can get an NE strategy.

If the offline dataset is the random dataset, according to the generation process of the random dataset and Assumption D.8, the behavior strategy used to generate the random dataset is a uniform strategy. Therefore, the behavior cloning algorithm can only get a uniform strategy instead of the equilibrium strategy under the random dataset. $\square$

Our experimental results also show the same outcomes as Theorem D.8. The performance of the behavior cloning technique mainly depends on the quality of the behavior strategy used to generate the offline dataset. Therefore, the behavior cloning technique can perform well under the expert dataset. Based on the above two theorems, we propose our OEF algorithm, BCMB, by combining the above two techniques with different weights to improve the performance under these datasets with unknown behavior strategies.

**Theorem D.9** (Theorem 5.3)**.** *Under the assumptions in Theorems D.7 and D.8, BCMB can compute the equilibrium strategy under either the random dataset or the expert dataset.*

*Proof.* In the BCMB algorithm, the weight of the BC policy is represented by $\alpha$. The weight of the MB policy is $1 - \alpha$. The $\alpha$ ranges from 0 to 1. When under the random dataset, let $\alpha$ equal 0. Then the policy of BCMB would equal to MB policy, i.e., the policy trained using the model-based algorithm. According to Theorem D.7, the model-based framework can converge to an equilibrium strategy under the random dataset. Therefore, BCMB can also converge to an equilibrium strategy under the random dataset.

When under the expert dataset, let $\alpha$ equal to 1. Then the policy of BCMB would be equal to BC policy, i.e., the policy trained by the behavior cloning algorithm. Similarly, according to Theorem D.8, BCMB can get an equilibrium strategy in the expert dataset. $\square$

Let's move to a more general case in which the offline dataset is generated by a behavior strategy $\sigma$. Then we have the following theorems under the general case.

**Theorem D.10.** *Assuming that the offline dataset $\mathcal{D}_\sigma$ generated by the behavior strategy $\sigma$ covers $(s_t, a, s_{t+1}), \forall s_t, a \in A(s_t)$ and the environment model is trained on $\mathcal{D}_\sigma$ with an extremely small training error $\epsilon$, the model-based framework can converge to an equilibrium strategy that performs equal even better than $\sigma$.*

*Proof.* According to the proof of Theorem D.7, since every state transition of the actual game is covered by $\mathcal{D}_\sigma$, the trained environment model would be the same as the actual game under the assumption that the environment model is trained on the offline dataset with an extremely small training error $\epsilon$. Then according to Theorem D.7, the model-based framework can converge to an equilibrium strategy. If $\sigma$ used to generate the dataset is not the equilibrium strategy, then the model-based framework can get a better strategy (equilibrium strategy) than $\sigma$. And if $\sigma$ is an equilibrium strategy, then the strategy trained using a model-based framework would perform equal to $\sigma$. $\square$

**Theorem D.11.** *Assuming that the behavior cloning policy is trained on the offline dataset $\mathcal{D}_\sigma$ generated by the behavior strategy $\sigma$ with an extremely small training error $\epsilon$, the performance of behavior cloning policy $\sigma^{bc}$ would be as good as the performance of $\sigma$.*

*Proof.* According to the Assumption D.8, behavior cloning can precisely mimic the behavior strategy in the offline dataset. Therefore, $\sigma^{bc}$ would be same as $\sigma$. Consequently, the performance of $\sigma^{bc}$ would have the same performance as $\sigma$. $\square$

**Theorem D.12** (Theorem 5.4)**.** *Assuming that the environment model and the behavior cloning policy are trained with an extremely small training error $\epsilon$ on the offline dataset $\mathcal{D}_\sigma$ generated using $\sigma$, BCMB can get an equal or better strategy than $\sigma$.*

*Proof.* Following the proof of Theorem D.9, let $\alpha$ equal 1. Then BCMB would reduce to BC. Then according to Theorem D.11, the performance of BC policy is at least as good as $\sigma$. Therefore, BCMB can get a strategy that is at least as good as the behavior strategy $\sigma$.

In another extreme case in which $\mathcal{D}_\sigma$ covers $(s_t, a, s_{t+1}), \forall s_t, a \in A(s_t)$, let $\alpha$ equal to 0. Then BCMB would reduce to MB. Then according to Theorem D.10, the MB policy performs equal to or better than $\sigma$. Therefore, in this case, BCMB can get an equal or better strategy than $\sigma$. $\square$

In conclusion, under the above assumptions, BCMB can perform at least equal to the behavior strategy used to generate the offline dataset. The improvement over the behavior strategy mainly depends on the performance of the model-based algorithm under the offline dataset.

# E   Implementation Details

**Model-based Framework.** Next, we introduce our instantiate offline model-based algorithms: OEF-PSRO and OEF-CFR, which are adaptions from two widely-used online equilibrium finding algorithms PSRO and Deep CFR, and OEF-JPSRO, which is an adaption from JPSRO. These three algorithms perform on the trained accurately environment model $E$. We first introduce the OEF-PSRO algorithm, and the whole flow is shown in Algorithm 2. Firstly, we need the trained accurately environment model $E$ as input and initialize policy sets $\Pi$ for all players using random strategies. Then, we need to estimate a meta-game matrix by computing expected utilities for each joint strategy profile $\pi \in \Pi$. In vanilla PSRO, to get the expected utility for $\pi$, it needs to perform the strategy $\pi$ in the actual game simulator. However, the simulator is missing in the OEF setting. Therefore, we use the trained accurately environment model $E$ to replace the game simulator to provide the information needed in the algorithm. Then we initialize meta-strategies using a uniform strategy. Next, we need to compute the best response policy oracle for every player and add the best response policy oracle to their policy sets. When training the best response policy oracle using DQN or other reinforcement learning algorithms, we sample the training data based on the environment model $E$. After that, we compute missing entries in the meta-game matrix and calculate meta-strategies for the meta-game. To calculate the meta-strategy $\sigma$ of the meta-game matrix, we can use the Nash solver or $\alpha$-rank algorithm. Here, we use the $\alpha$-rank algorithm as the meta solver because our algorithm needs to solve multi-player games. Finally, we repeat the above process until satisfying the convergence conditions. Since the process of JPSRO is similar to PSRO except for the best response computation and meta distribution solver, OEF-JPSRO is also similar to OEF-PSRO. We do not cover OEF-JPSRO in detail here.

---

**Algorithm 2** Offline Equilibrium Finding - Policy-Space Response Oracles

---

1: **Input:** Trained environment model $E$
2: Initial policy sets $\Pi$ for all players
3: Compute expected utilities $U^\Pi$ for each joint $\pi \in \Pi$ **based on the environment model** $E$
4: Initialize mate-strategies $\sigma_i = \text{UNIFORM}(\Pi_i)$
5: **repeat**
6:     **for** player $i \in [1, .., n]$ **do**
7:         **for** best response episodes $p \in [0, ..., t]$ **do**
8:             Sample $\pi_{-i} \sim \sigma_{-i}$
9:             Train best response $\pi_i'$ over $\rho \sim (\pi_i', \pi_{-i})$, which **samples on the environment model** $E$
10:         **end for**
11:         Compute missing entries in $U^\Pi$ from $\Pi$ **based on the environment model** $E$
12:         Compute a meta-strategy $\sigma$ from $U^\Pi$ using $\alpha$-rank algorithm;
13:     **end for**
14: **until** Meet convergence condition
15: **Output:**current solution strategy $\sigma_i$ for player $i$

---

Algorithm 3 shows the process of OEF-CFR. It also needs the trained environment model $E$ as input. We first initialize regret and strategy networks for every player and then initialize regret and strategy memories for every player. Then we need to update the regret network for every player. To do this, we can perform the traverse function to collect corresponding training data. The traverse function can be any sampling-based CFR algorithm. Here, we use the external sampling algorithm. Note that we need to perform the traverse function on the game tree. In OEF-CFR, the trained environment model can replace the game tree. Therefore, the trained environment model is the input of the traverse function. Algorithm 4 shows the process of the traverse function. In this traverse function, we collect the regret training data of the traveler, and the strategy training data of other players are also gathered. After performing the traverse function several times, the regret network is updated using the regret memory. We need to repeat the above processes $n$ iterations. Then the average strategy network for every player is trained based on its corresponding strategy memory. Finally, the trained average strategy networks are output as the approximate NE strategy.

**Combination Method.** As introduced in the main paper, we proposed three methods to select the parameter for combination. Here, we detailly introduce the method that uses a parameter predictor since the

---

**Algorithm 3** Offline Equilibrium Finding - Deep Counterfactual Regret Minimization

---

1: **Input:** Trained environment model $E$
2: Initialize regret network $R(I, a|\theta_{r,p})$ for every player $p$;
3: Initialize average strategy network $S(I, a|\theta_{\pi,p})$ for every player $p$;
4: Initialize regret memory $M_{r,p}$ and strategy memory $M_{\pi,p}$ for every player $p$;
5: **for** CFR Iteration $t = 1$ to $T$ **do**
6:    **for** player $p \in [1, ..., n]$ **do**
7:        **for** traverse episodes $k \in [1, ..., K]$ **do**
8:            TRVERSE($\phi$, $p$, $\theta_{r,p}$, $\theta_{\pi,-p}$, $M_{r,p}$, $M_{\pi,-p}$, $E$);
9:        **end for**
10:        Train $\theta_{r,p}$ from scratch based on regret memory $M_{r,p}$
11:    **end for**
12: **end for**
13: Train $\theta_{\pi,p}$ based on strategy memory $M_{\pi,p}$ for every player $p$;
14: **Output:**$\theta_{\pi,p}$ for every player $p$

---

other two methods are straightforward. Firstly, we need to collect the training data by getting the best parameters through online interaction for different offline datasets. We use CKA to measure the difference between the bc policy and mb policy since these two policies are both neural networks. Therefore, the differences between each layer of these two policies get from different offline datasets are taken as the input and the best parameter for different offline datasets is taken as the output. Then we can easily use these data to train a parameter predictor. Finally, when encountering a new OEF problem, we can directly get the parameter from the parameter predictor based on the difference between the trained bc policy and the trained mb policy. This parameter predictor can also be used for other games. Here, we conduct some experiments to show the performance of the proposed combination method. Figures 15 and 16 show these experimental results. We can find that BC+MB can provide a good parameter for the unseen game while the performance depends on the difference between the unseen game and the game used to train the parameter predictor.

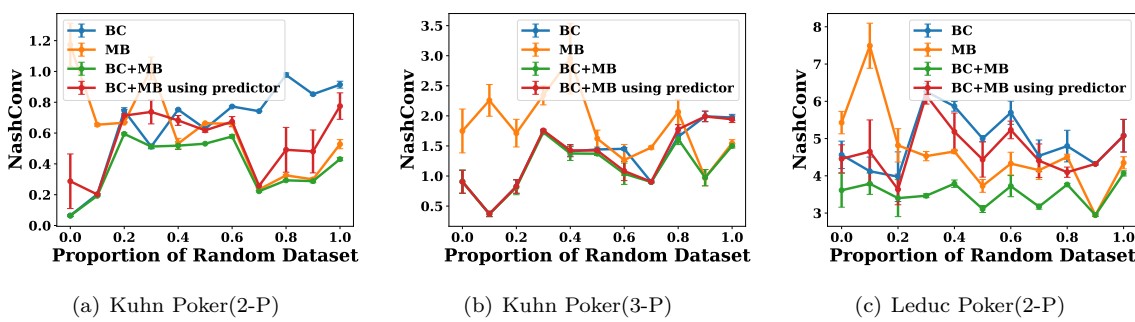

(a) Kuhn Poker(2-P)          (b) Kuhn Poker(3-P)          (c) Leduc Poker(2-P)

Figure 15: Experimental results for different combination methods (parameter predictor trained on two-player kuhn poker)

---

**Algorithm 4** TRVERSE($s, p, \theta_{r,p}, \theta_{\pi,-p}, M_{r,p}, M_{\pi,-p}, E$)-External Sampling Algorithm

---

1: **if** $s$ is terminal state **then**
2:     Get the utility $u_i(s)$ from environment model $E$
3:     **Output:** $u_i(s)$
4: **else**
5:     **if** $s$ is a chance state **then**
6:         Sample an action $a$ based on the probability $\sigma_c(s)$, which is obtained from model $E$
7:         $s' = E(s, a)$
8:         **Output:** TRAVERSE($s', p, \theta_{r,p}, \theta_{\pi,-p}, M_{r,p}, M_{\pi,-p}, E$)
9:     **else**
10:         **if** $P(s) = p$ **then**
11:             $I \leftarrow s[p]$; # game state is formed by information sets of every player
12:             $\sigma(I) \leftarrow$ strategy of $I$ computed using regret values $R(I, a|\theta_{r,p})$ based on regret matching
13:             **for** $a \in A(s)$ **do**
14:                 $s' = E(s, a)$
15:                 $u(a) \leftarrow$ TRAVERSE($s', p, \theta_{r,p}, \theta_{\pi,-p}, M_{r,p}, M_{\pi,-p}, E$)
16:             **end for**
17:             $u_\sigma \leftarrow \sum_{a \in A(s)} \sigma(I, a) u(a)$
18:             **for** $a \in A(s)$ **do**
19:                 $r(I, a) \leftarrow u(a) - u_\sigma$
20:             **end for**
21:             Insert the infoset and its action regret values $(I, r(I))$ into regret memory $M_{r,p}$
22:             **Output:** $u_\sigma$
23:         **else**
24:             $I \leftarrow s[p]$
25:             $\sigma(s) \leftarrow$ strategy of $I$ computed using regret value $R(I, a|\theta_{r,-p})$ based on regret matching
26:             Insert the infoset and its strategy $(I, \sigma(s))$ into strategy memory $M_{\pi,-p}$
27:             Sample an action $a$ from the probability distribution $\sigma(s)$
28:             $s' = E(s, a)$
29:             **Output:** TRAVERSE($s', p, \theta_{r,p}, \theta_{\pi,-p}, M_{r,p}, M_{\pi,-p}, E$)
30:         **end if**
31:     **end if**
32: **end if**

---

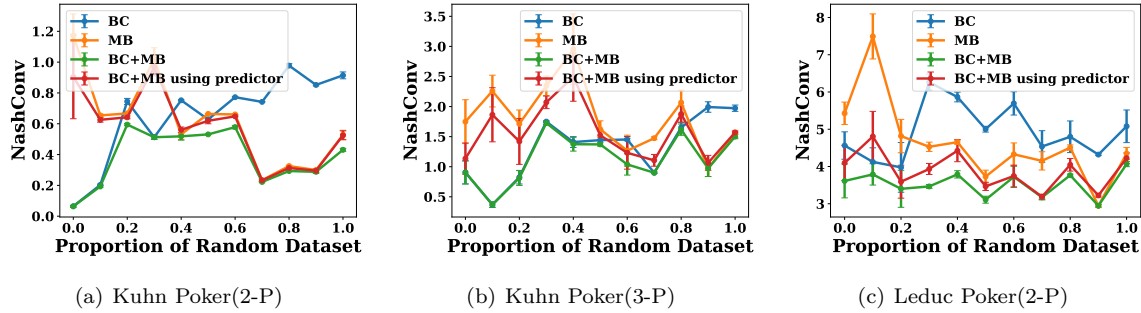

(a) Kuhn Poker(2-P)          (b) Kuhn Poker(3-P)          (c) Leduc Poker(2-P)

Figure 16: Experimental results for different combination methods (parameter predictor trained on two-player leduc poker)

## F   Additional Experimental Results

In this section, we provide experimental results on other games. First, we present the experimental results of the behavior cloning method and model-based framework (OEF-CFR) using hybrid datasets, followed by the results of our OEF algorithm (BC+MC). We also test our OEF algorithm on a two-player Phantom Tic-Tac-Toe game using the learning dataset. Finally, we provide an ablation study and the setting of hyper-parameters used in our experiments.

Figure 17 displays the results of the behavior cloning technique on several multi-player poker games and a two-player Liar's Dice game. The results show that as the proportion of random datasets increases, performance decreases. This observation is consistent with the finding from our previous experiments.

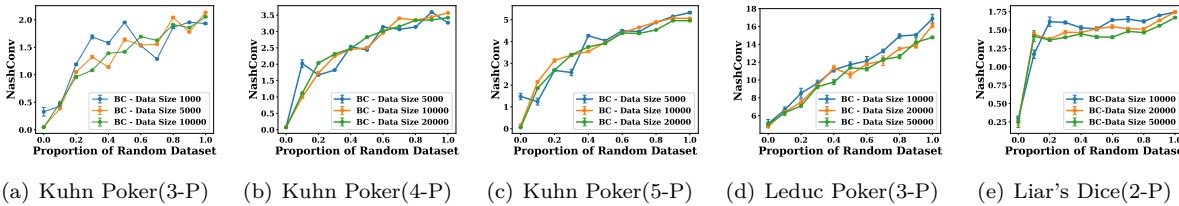

(a) Kuhn Poker(3-P)   (b) Kuhn Poker(4-P)   (c) Kuhn Poker(5-P)   (d) Leduc Poker(3-P)   (e) Liar's Dice(2-P)

Figure 17: Experimental results for the BC method

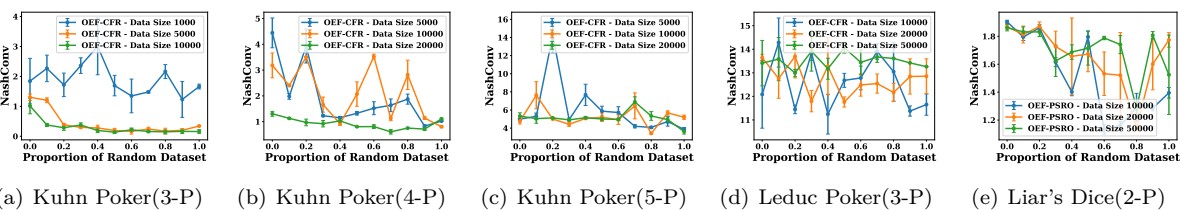

(a) Kuhn Poker(3-P)   (b) Kuhn Poker(4-P)   (c) Kuhn Poker(5-P)   (d) Leduc Poker(3-P)   (e) Liar's Dice(2-P)

Figure 18: Experimental results for the MB method

The experimental results of the model-based framework (OEF-CFR) on these games are shown in Figure 18. Since the strategy learned by OEF-CFR is not a joint strategy, we only use NashConv to measure its closeness to NEs in these multiple-player games. From these results, we find that the performance of the model-based framework is not stable in these games but still shows a slightly decreasing tendency with the increase in the proportion of the random dataset. Note that the CFR-based algorithm has no theoretical guarantee of convergence in multiple-player games. Therefore, OEF-CFR also cannot guarantee convergence to the NE strategy. The performance of the model-based framework also depends on the trained environment model. As a result, poor performance may be caused by an inadequately trained environment model or the poor performance of the CFR-based algorithm in multiple-player games. Hence, learning a good enough strategy is a significant challenge in these multiple-player games under the OEF setting.

Figures 19(a)-19(j) show the experimental results of BCMB on various games. We also test our OEF algorithm BCMB in the Phantom Tic-Tac-Toe game based on the learning dataset (Figure 19(k)). The NashConv values in Phantom Tic-Tac-Toe are approximate results since the best response policy is trained using DQN, and the utilities are obtained by simulation. The results show that the BCMB performs better than BC and MB, which implies that our combination method can perform well in any game under any unknown dataset. The appropriate weights in the BCMB algorithm under different datasets are shown in Figure 20. This displays a similar tendency as in previous experiments.

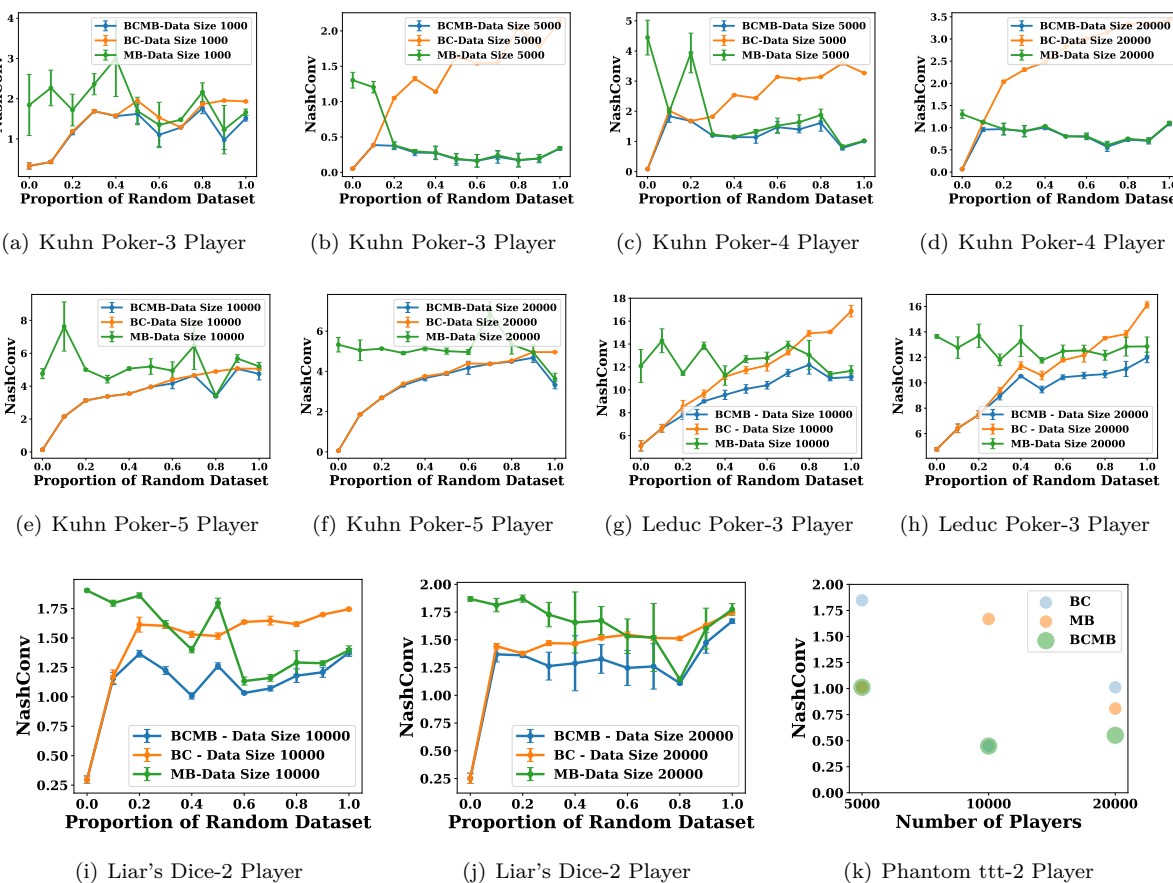

Figure 19: Experimental results for the benchmark algorithm BCMB

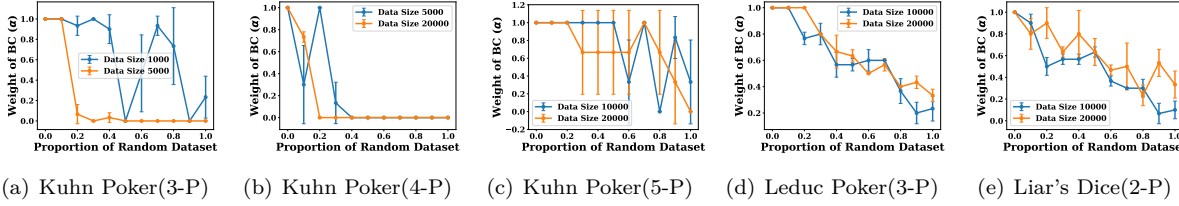

Figure 20: Experimental Results for Proper Weight

**Ablation Study.** To investigate the influence of hyperparameters, we conduct several ablation experiments on two-player Kuhn poker and Leduc poker games. We consider different model structures with various numbers of hidden layers. Specifically, for the 2-Player Kuhn poker game, we use different environment models with 8, 16, 32, and 64 hidden layers. For the 2-Player Leduc poker game, which is a more complicated game, the numbers of hidden layers for different models are 32, 64, and 128. In addition, we train the environment models for different epochs to evaluate the robustness of our approach. Figures 21-22 show these ablation results. We find that the number of hidden layers and the number of training epochs have little effect on the performance of the BC algorithm. These results further verify that the performance of the BC algorithm primarily depends on the quality of the dataset. As we know, the performance of the model-based framework mainly depends on the trained environment model. Since the number of the hidden layer and the number of training epochs influence the training phase of the environment model, the number of the hidden layer and the number of train epochs have a slight impact on the performance of the model-based framework. As long as the size of the hidden layer and the number of training epochs can guarantee that the environment model is trained accurately, the performance of the model-based framework will not be affected.

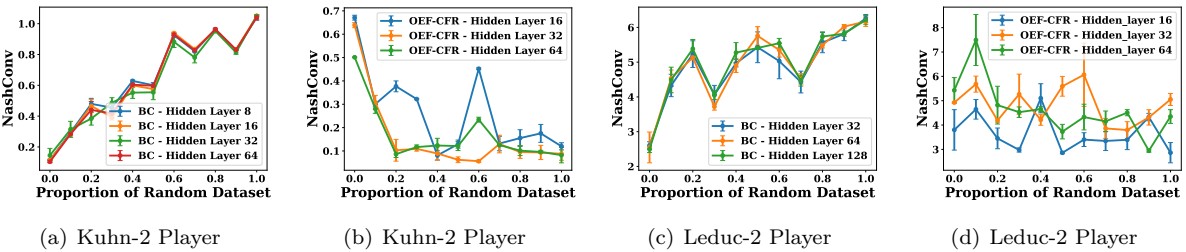

(a) Kuhn-2 Player     (b) Kuhn-2 Player     (c) Leduc-2 Player     (d) Leduc-2 Player

Figure 21: Ablation results for different hidden layer size

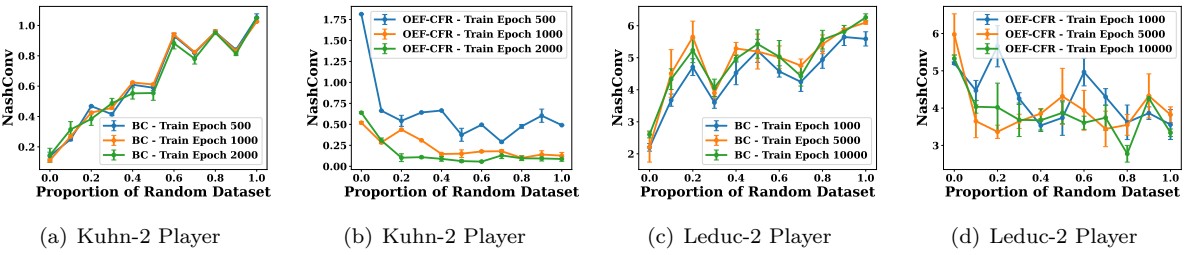

(a) Kuhn-2 Player     (b) Kuhn-2 Player     (c) Leduc-2 Player     (d) Leduc-2 Player

Figure 22: Ablation results for different train epoch

**Parameter Setting.** We list the parameters used to train the behavior cloning policy and environment model for all games used in our experiments in Table 2 and Table 3.

Table 2: Parameters for Behavior Cloning algorithm

| Games | Data size | Hidden layer | Batch size | Train epoch |
|---|---|---|---|---|
| 2-player Kuhn poker | 500 | 32 | 32 | 1000 |
| 2-player Kuhn poker | 1000 | 32 | 32 | 2000 |
| 2-player Kuhn poker | 5000 | 32 | 32 | 2000 |
| 3-player Kuhn poker | 1000 | 32 | 32 | 5000 |
| 3-player Kuhn poker | 5000 | 32 | 32 | 5000 |
| 3-player Kuhn poker | 10000 | 64 | 128 | 5000 |
| 4-player Kuhn poker | 5000 | 64 | 64 | 5000 |
| 4-player Kuhn poker | 10000 | 64 | 128 | 5000 |
| 4-player Kuhn poker | 20000 | 64 | 128 | 5000 |
| 5-player Kuhn poker | 5000 | 64 | 64 | 5000 |
| 5-player Kuhn poker | 10000 | 64 | 128 | 5000 |
| 5-player Kuhn poker | 20000 | 64 | 128 | 5000 |
| 2-player Leduc poker | 10000 | 128 | 128 | 10000 |
| 2-player Leduc poker | 20000 | 128 | 128 | 10000 |
| 2-player Leduc poker | 50000 | 128 | 128 | 10000 |
| 3-player Leduc poker | 10000 | 128 | 128 | 10000 |
| 3-player Leduc poker | 20000 | 128 | 128 | 10000 |
| 3-player Leduc poker | 50000 | 128 | 128 | 10000 |
| Liar's Dice | 10000 | 64 | 64 | 5000 |
| Liar's Dice | 20000 | 64 | 128 | 5000 |
| Liar's Dice | 50000 | 64 | 128 | 5000 |
| Phantom Tic-Tac-Toe | 5000 | 128 | 128 | 5000 |
| Phantom Tic-Tac-Toe | 10000 | 128 | 128 | 5000 |
| Phantom Tic-Tac-Toe | 20000 | 128 | 128 | 5000 |

Table 3: Parameters for training Environment Model

| Games | Data size | Hidden layer | Batch size | Train epoch |
|---|---|---|---|---|
| 2-player Kuhn poker | 500 | 32 | 32 | 1000 |
| 2-player Kuhn poker | 1000 | 32 | 32 | 2000 |
| 2-player Kuhn poker | 5000 | 32 | 32 | 2000 |
| 3-player Kuhn poker | 1000 | 32 | 32 | 2000 |
| 3-player Kuhn poker | 5000 | 32 | 32 | 5000 |
| 3-player Kuhn poker | 10000 | 64 | 128 | 5000 |
| 4-player Kuhn poker | 5000 | 64 | 64 | 5000 |
| 4-player Kuhn poker | 10000 | 64 | 128 | 5000 |
| 4-player Kuhn poker | 20000 | 64 | 128 | 5000 |
| 5-player Kuhn poker | 5000 | 64 | 64 | 5000 |
| 5-player Kuhn poker | 10000 | 64 | 128 | 5000 |
| 5-player Kuhn poker | 20000 | 64 | 128 | 5000 |
| 2-player Leduc poker | 5000 | 64 | 64 | 5000 |
| 2-player Leduc poker | 10000 | 64 | 64 | 5000 |
| 2-player Leduc poker | 20000 | 128 | 128 | 10000 |
| 3-player Leduc poker | 10000 | 128 | 128 | 10000 |
| 3-player Leduc poker | 20000 | 128 | 128 | 10000 |
| 3-player Leduc poker | 50000 | 128 | 128 | 10000 |
| Liar's Dice | 10000 | 64 | 64 | 5000 |
| Liar's Dice | 20000 | 64 | 128 | 5000 |
| Liar's Dice | 50000 | 64 | 128 | 5000 |
| Phantom Tic-Tac-Toe | 5000 | 128 | 128 | 5000 |
| Phantom Tic-Tac-Toe | 10000 | 128 | 128 | 5000 |
| Phantom Tic-Tac-Toe | 20000 | 128 | 128 | 5000 |

