# OpenReview forum: "Offline Equilibrium Finding"
_TMLR — Rejected by TMLR_

### Review · Reviewer_FpMY · 2023-05-14

**Summary Of Contributions:**

This paper addresses the problem of learning equilibrium strategies in $n$-players imperfect-information extensive-form games relying on an offline dataset. First, the paper introduces a corresponding learning framework, called Offline Equilibrium Finding (OEF), which instantiate the problem of extracting either a Nash or correlated equilibrium from an offline dataset. Then, it presents a methodology (BCMB) that combines behavioral cloning with a model-based procedure employing popular game-solving algorithms (e.g., PSRO and CFR) on the approximated model. Finally, the paper provides some theoretical guarantees, which tie the quality of the data to the quality of the solution, as well as an empirical validation in simple domains.

**Audience:**

Yes

**Broader Impact Concerns:**

This paper can be categorized as fundamental research. I believe explicitly commenting on the potential negative impacts is not necessary at this stage of development.

**Claims And Evidence:**

No

**Requested Changes:**

1) (Major, Novelty) The paper shall explicitly comment the novelty of the contribution w.r.t. prior work (e.g., Cui & Du, 2022).

2) (Minor, Presentation) Report the definition of (coarse) correlated equilibria in mathematical terms.

3) (Minor, Presentation) It is hard to understand how CFR and PSRO work from the provided description. I would rather extend the presentation of the algorithms or avoid the description entirely (just pointing to the references instead).

4) (Major, Finite-sample analysis) Most of the theoretical statements start with "Assume that the environment model and/or the behavioral cloning policy are trained accurately". On the one hand, this assumption is ambiguous, as it is unclear whether the authors mean the estimates are "accurate" given the dataset (aka training accuracy) or in general (aka generalization accuracy). In the latter case, the assumption is reasonable only assuming infinite datasets. Typically, the crucial challenge of (theoretical) offline learning is to provide a guarantee on the solution given the quality (more on this below) **and** the size of the dataset.

5) (Major, Coverage assumptions) The coverage assumptions are not stated formally. While the Ass. 5.1 is somewhat clear, the 5.2 is not.

6) (Major, Model-free vs model-based) The discussion in the initial paragraphs of Sec. 5 seem to imply that model-free offline learning cannot work in OEF, and model-based methods has to be considered (at least in combination with other model-free). This claim does not seem to be supported, especially if we do not restrict model-free methods to BC. In offline RL there exists several model-free methods with guarantees, and I am wondering whether it cannot be the case in OEF as well.

**Strengths And Weaknesses:**

Strengths
- The paper tackles the very relevant problem of solving games with offline datasets;
- The paper is well-organized and easy to read.

Weaknesses
- Claims, definitions, assumptions are not formal enough (e.g., "trained accurately" or "covers the NE strategy" are not translated in mathematical terms);
- I am not familiar with the literature of learning in games, but the novelty of this work is not sufficiently supported in the paper, and previous works seem to introduce similar formulations (e.g., Cui & Du, 2022);
- The paper reports some asymptotic "possibility" results that are not particularly surprising (if I am understanding them correctly), and does not include a finite-samples analysis.

To my understanding, the main contribution of this work is an heuristic that performs well with "hybrid" datasets, as it combines BC, which generally works better with experts' data, together with a model-based procedure, which works better with random data, from which a model of the game can be estimated. While all of this sounds reasonable, I think the paper lacks either a strong theoretical support for the introduced method, or a convincing experimental analysis that compare BCMB against competitive baselines in complex domains. Without those, I am wondering whether the contribution is sufficient to warrant publication.

---

> ### Author Response · Authors · 2023-06-16
> **Response to Reviewer FpMY**
>
> **Q1: The paper shall explicitly comment the novelty of the contribution w.r.t. prior work (e.g., Cui & Du, 2022).**
>
> A1:  We summarize our contribution with respect to Cui & Du's 2022 work as follows:
> While prior research, including that of Cui & Du, has primarily aimed at solving Markov games, our objective is to address extensive-form games. Specifically, Cui & Du's work concentrates on computing the Nash equilibrium strategy for tabular two-player zero-sum Markov games. In contrast, our study expands on this by not only computing the Nash equilibrium strategy for two-player zero-sum extensive-form games, but also devising the Correlated (or Coarse Correlated) Equilibrium for multi-player extensive-form games. Moreover, while their theoretical findings focus exclusively on two-player zero-sum Markov games, our work broadens these results to incorporate the extensive-form game setting.
>
> The algorithms for solving Extensive-form games and Markov games generally differ, mainly due to the unique structure and properties of each game type. Markov games focus more on state transitions and Markov properties of strategies, thus often employing reinforcement learning-based algorithms. With the success of offline RL, its principles can be transferred to offline learning in Markov games. On the other hand, extensive-form games emphasize the sequence of decisions over time and the presence of imperfect information. Therefore, algorithms typically used to solve extensive-form games include the linear programming-based algorithm, the Counterfactual regret minimization algorithm, and the Policy Space Response Oracle algorithm. Given these characteristics, solving the extensive-form games in an offline setting presents a more formidable challenge compare to Markov games.
>
> We appreciate your insightful suggestion. We have added these comparisons to the main paper to highlight our contributions and the differences between our work and prior work (Introduction Section). Thank you for helping enhance the clarity of our paper.
>
> **Q2: Claims, definitions, assumptions are not formal enough and Report the definition of (coarse) correlated equilibria in mathematical terms.**
>
> A2: We appreciate your review and feedback. The theoretical results in Section 5.4 have been revised and the formal definitions of the (coarse) correlated equilibria have been included in Section 2.1.
>
> **Q3:  It is hard to understand how CFR and PSRO work from the provided description.**
>
> A3: Thank you for your feedback. We have updated the descriptions for the Counterfactual Regret Minimization (CFR) and Policy Space Response Oracle (PSRO) algorithms in Section 2.2.
>
> **Q4: (Finite-sample analysis) Most of the theoretical statements start with "Assume that the environment model and/or the behavioral cloning policy are trained accurately". On the one hand, this assumption is ambiguous, as it is unclear whether the authors mean the estimates are "accurate" given the dataset (aka training accuracy) or in general (aka generalization accuracy). In the latter case, the assumption is reasonable only assuming infinite datasets. Typically, the crucial challenge of (theoretical) offline learning is to provide a guarantee on the solution given the quality (more on this below) and the size of the dataset.**
>
> A4: We appreciate your review. In response to your feedback, we have revised any ambiguous wording and supplemented the Appendix with some sample analyses. To be specific, since the behavior cloning policy and the environment model are both trained in a supervised learning manner, we provide a general generalization bound for training such neural network models.
>
> **Q5:  The coverage assumptions are not stated formally. While the Ass. 5.1 is somewhat clear, the 5.2 is not.**
>
> A5: Thank you for your feedback. We have updated the description for Assumption 5.2 in Section 5.4 based on your review.

---

> > ### Comment · Reviewer_FpMY · 2023-06-27
> > **After Response**
> >
> > I want to thank the authors for their detailed response. I think they are making a good point in noting that previous works only focused on offline learning in Markov games rather than extensive-form. Not being an expert in the field, I cannot uphold this claim in principle, but this would make the OEF problem way more relevant than I originally thought.
> >
> > However, I think the authors should keep working on
> > - improving the quality of the theoretical statements and results
> > - clarifying the issue raised by Reviewer Patg on the online tuning of $\alpha$
> >
> > to clear the bar for acceptance.

---

> > > ### Author Response · Authors · 2023-06-30
> > > **Response to Reviewer FpMY**
> > >
> > > Thanks for your responses. For your concerns, we have already made big modifications.
> > >
> > > 1.	For the theoretical results, we have summarized our main results in Section 5.4 and more theoretical analyses in Appendix D. We first provide some theoretical analysis for solving the OEF problem by comparing it with offline RL. Then we provide a bound for training the neural network we used in our method. Finally, we provide the performance guarantee under different offline datasets for each algorithm.
> > >
> > > 2.	For the selection of $\alpha$, we have proposed three methods. For the fully online method, since we cannot get any feedback from the actual environment, we can only randomly select a $\alpha$ and set it constantly when using. If we can get some feedback from the actual environment, then we can use our search method to select the optimal $\alpha$. However, this method may need many interactions with the actual environment. To this end, we propose the third method – training a neural network to predict the optimal $\alpha$ based on the difference between the BC policy and MB policy. In this way, we only need some feedback from the actual environment to train the neural network. Then we can use it in different cases and even other similar games. We have provided these methods in Section 5.3 and some experiment results in Appendix E.

---

> > > > ### Comment · Reviewer_FpMY · 2023-06-30
> > > > **Discussion**
> > > >
> > > > Thank you for the further clarifications and for updating the paper. Unfortunately, I do not think those changes are sufficient.
> > > >
> > > > For the theoretical statements, I am worried that the problem may be not limited to how the statement are presented, but how they are derived.
> > > >
> > > > For the second point, I appreciate the discussion of the authors on how to overcome such issue, but this still reduces the value of the proposed approach.

---

> ### Author Response · Authors · 2023-06-16
> **Response to Reviewer FpMY (Cont.)**
>
> **Q6: The discussion in the initial paragraphs of Sec. 5 seem to imply that model-free offline learning cannot work in OEF, and model-based methods has to be considered (at least in combination with other model-free). This claim does not seem to be supported, especially if we do not restrict model-free methods to BC. In offline RL there exists several model-free methods with guarantees, and I am wondering whether it cannot be the case in OEF as well.**
>
> A6: The primary reason model-free methods aren't suitable for Offline Extensive-Form (OEF) games is that calculating the Nash Equilibrium (NE) strategy for such games is a minimax optimization problem, not a simple optimization problem. Model-free methods, which primarily focus on solving single maximum or minimum optimization problems, are not easily adaptable to compute the NE strategy, which requires addressing both maximum and minimum conditions simultaneously. Therefore, the model-free offline RL method cannot guarantee to converge to the NE strategy.
>
> **Q7:  I think the paper lacks either a strong theoretical support for the introduced method, or a convincing experimental analysis that compare BCMB against competitive baselines in complex domains.**
>
> A7: We appreciate your review. In this paper, we are the first to introduce the OEF setting for solving the extensive-form game in an offline manner. To facilitate comprehension of the OEF problem and the OEF algorithm, we have conducted theoretical analyses that encompass the minimal dataset assumption for addressing the OEF problem, as well as sample analysis and performance guarantee for our OEF algorithm. Although we have not carried out experiments in complex domains, we have performed extensive experiments across various games to evaluate the performance of the OEF algorithm. As the first study to solve the extensive-form game offline using an empirical algorithm, we selected several popular extensive-form games as our testing platforms. We hope that our work can shed light on how to tackle real-world scenarios in an offline setting.

---

### Review · Reviewer_Patg · 2023-05-14

**Summary Of Contributions:**

This paper tackles the Offline Equilibrium Finding problem: given a dataset of transitions in a multiplayer game, find the Nash Equilibrium (or the (coarse) correlated equilibruim) of the game.

To do that, they first build interaction dataset on standard benchmarks game, in a similar manner datatsets for offline RL have been constructed, ie recording interactions in the game with different polices: random, expert, or a mix of both.

Then, they design an algorithm to solve OEF, they call BCMB. It is a mixture between a model-based method and behavior cloning on the dataset. The model-based method is adapted form an online equilibrium-finding algorithm: it learns a model from the dataset, and then runs an online algorithm on the dataset.

The finally test their method, compared with offline RL baselines, on the OpenSpiel benchmark.

**Audience:**

Yes

**Broader Impact Concerns:**

No broader impact statement is needed.

**Claims And Evidence:**

No

**Requested Changes:**

To explain the requested changes and the conclusion of my review: my background is mainly in RL, and I am not an expert on game theory, thus I will stick to the RL and experimental part for the key parts of the decision.

**Selection of $\alpha$**

I think the main issue of the paper is the online selection of the mixture (Weakness 1). As of now,  I think the claim that "BCMB is better than offline RL to find a NE" is not really supported. What could change:
 - maybe all the other baseline are also finetuned online on the game, in that case the issue is not with this work in particular
- at least the wording should be different, and this limitation should be clearly stated and discussed.
- the best thing would be to have a way to select alpha offline (or online, but the same for every dataset), but this may be a hard problem in itself

**Theory and assumptions**

Then, I think revisiting the presentation of the theory would improve the paper (see weakness 2):
- I would suggest discussing more the relation of the th 5.2 and 5.3 to the assumptions, and the realstic aspect of the assumptions, in the main text of the paper.

**Strengths And Weaknesses:**

# Strengths

The paper is generally well written, well organised and easy to follow. Its major strengths are the clarity of its contributions, detailed below.

## New setting

OEF is an interesting and novel setting. Offline RL has had a growing interest in the recent years, I think it is a important direction to also tackle this setting from a game theory perspective.

## Simple ideas

The MB method is a straightforward adaption of online methods to the offline setting. It is quite interesting to know if this works or not in practice.

## Data collection

 Authors have built datasets for OEF. If there are published, this is a useful contribution to the community.

# Weaknesses

I have a few concerns about the paper, not all of the same importance. "major weaknesses" are what I think should be taken into account for the acceptance, "minor" are easily fixed.

## Major weakness 1: online parameter selection

The main weakness of this work is that BCMB is actually not an offline algorithm Precisely, the $\alpha$ parameter that controls the mixture between BC and MB is selected by a grid search *online, on the environment*.  This causes a few issues:

 First, it breaks the assumption that the algorithms can compute a policy without access to the environment. Without this finetuning, this paper does not provide a way to select $\alpha$ offline. This is a known issue in imitation and offline learning (for example see [1, 2]). Having environment-specific parameters is understandable in some cases, but here it depends on the dataset of the same env (whether it is random collection or expert). This is not just a question of over-tuning HPs. The mixture coefficient is a key component of the algorithm, and needs to depend on the quality of the policy that collected the data; but knowing this quality from offline interaction is a problem by itself.

The comparison to the offline RL algorithm is not really fair. IIUC, they use the same set of HPs for each game, and are not finetuned online on each dataset.

Overall, this is a strong limitation of the method, that is not mentioned or discussed. BCMB can provide insights on what is necessary for OEF, but right now it is a but misleading to present it as outperforming purely offline methods.

[1]: Hyperparameter Selection for Imitation Learning, Hussenot et al, 2021

[2]: Hyperparameter Selection for Offline Reinforcement Learning, Le Paine et al, 2020


## Major weakness 2: relevance of the theory

One things that lacks some clarity to me is how the theory is presented in the work. especially theorems 5.2 and 5.3.

First, Theorem 5.2 is not really discussed when it is introduced, so in section 5.2, we do not really know how relevant it is, what is says precisely on the method, if the assumptions can be realistically respected, etc... For example: the results depends on the coverage assumption (assumption 5.1), but it is not clear if this assumption can realistically be realized. Usually, finding a covering dataset can be a hard exploration problem, and standard "random" datasets in offline RL becnhmarks (e.g., [3]) are generated by a uniform policy, and do not have this property.

Second, they seem to be direct consequences of assumptions (especially 5.3 which essentially says "we can put $\alpha=0$ or $\alpha=1$), I am not sure they deserve to be 'theorems'. This is not just a nomenclature issue: Th. 5.3 seems like it is backing up the algorithm with a convergence result, but it just says that form the mixture you can recover one of the element, I think it is a bit misleading.

Third, and this is related to the first weakness: I actually have an issue with the proof of theorem 3. The proof can be summarized as:
"if the dataset is random, choose $\alpha=0$, if expert, choose $\alpha=1$".
This exposes the limitation of the algorithm in weakness 1. Sure, these values of alpha are indeed optimal for this setting, but the algorithm has now way of finding them without access to the simulator. So, in the fully offline setting, BCMB cannot guarantee to be optimal, even under these assumption, because it cannot select the mixture value.

[3] D4RL: Datasets for Deep Data-Driven Reinforcement Learning, Fu et al, 2020

## Minor weaknesses

### Page limit

Some useful and interesting content is relegated to the appendix: the related work (which is quite complete and pedagogical), discussion of the theoretical analysis, in particular wrt the different coverage assumptions. It is not that much, but it feels a bit like playing on the appendix separation to get under the page limit, knowing that TMLR accepts longer articles.

### Details on figures

- bars on Figure 6 are overlapping and hard to read.
- colors are not consistent in Figures 7 and 8

---

> ### Author Response · Authors · 2023-06-16
> **Response to Reviewer Patg**
>
> **Q1: Selection of $\alpha$**
>
> A1: The mixed method indeed requires some level of online interaction. For clarity, let's delve deeper into the selection process of $\alpha$, outlining several methods for its selection.
>
> Given an offline dataset, we can get two strategies derived from offline datasets at our disposal, the most straightforward way is to randomly select the $\alpha$ between zero and one and maintain this $\alpha$ constant to get the mixed strategy. This method, however, doesn’t guarantee the optimal performance.
> To get the optimal $\alpha$, we advocate for the online method as detailed in the main paper. This method does necessitate some online interactions.
> A potential solution to this challenge involves training a neural network model to predict the optimal $\alpha$. This prediction would be based on the similarity between the behavior cloning strategy and the model-based strategy, as explained in Appendix E. While this solution might also require online interactions initially, the trained model can be reused across different games. Performance may vary, especially if a game substantially differs from the one used during the model's training phase. Nonetheless, it can be a remedial method for selecting $\alpha$ in an offline manner.
>
> **Q2: Theory and assumptions**
>
> A2: We appreciate your feedback. The theoretical sections in Section 5.4 and the Appendix have been updated and refined accordingly.

---

### Review · Reviewer_fZuh · 2023-06-03

**Summary Of Contributions:**

In short, the paper studies the problem of offline decision-making in games. The goal is to use historical data of gameplay to figure out equilibrium strategies for game playing. To do so, the paper proposes a set of datasets (based on the convention of random, hybrid and expert datasets common in offline RL and also logically motivated by game playing specifically) and then proposes a new model-based offline learning algorithm and a model-based + model-free learning algorithm to solve this problem.

**Audience:**

Yes

**Broader Impact Concerns:**

No broader impact statement in the paper. But I don't think this paper particularly has issues of broader impact.

**Claims And Evidence:**

Yes

**Requested Changes:**

I would request the authors to amplify the number of experiments and baselines in the experimental section (as discussed above), discuss all relevant related work, especially those that study the question of offline RL in Markov games, and finally fix the theoretical results. Without these, I don't think the paper is ready to be accepted to TMLR.

**Strengths And Weaknesses:**

Strengths:

- The paper studies an interesting problem setting, that has not been studied -- though I should say that theoretical RL literature does study offline RL in two-player games. See for example: https://arxiv.org/abs/2302.02571 and other related line of work.

Weaknesses:

- The main weakness of this paper comes from the fact that it does not actually compare to all the adequate baselines. While I do understand why model-free algorithms are not actually ideal when it comes to learning in games, however, it is not quite clear to me if they would necessarily be worse for the Hybrid dataset (my rationale is that if the dataset is near an equilibrium, the model of the environment and optimizing for the other opponent may not be needed as much, which is also why BC can work intuitively with the expert dataset). So not comparing model-free algorithms at all is a big weakness in my opinion. In fact, some more detailed studies should be done about it.

- Assumptions 5.1 and 5.2 are actually very strong -- in some ways, it does assume coverage of every state and action, with a non-stochastic reward function, which is not great. Even when typical offline RL works assume the setting of full coverage, they would actually try to analyze the setting where the reward function is stochastic and would account for error due to stochasticity despite full coverage. But I don't think the analysis in this paper accounts for that. All the theoretical results are very informal -- it is just explained in words, the meanings of "accurate", "error is low", etc are not explained. So I think the theory needs a full revision to improve rigor and clarity. Right now it is just hand wavy.

- How does the error in learning the model of the environment from offline data affect equilibrium finding? Does it need pessimism like most of offline RL? How does BCMB compare to standard model-based learning + CFR / JPSRO + pessimism? If pessimism is already imposed by the model-based algorithm, why do we need BC policy mixing? These questions need to be rigorously studied empirically and theoretically.

---

> ### Author Response · Authors · 2023-06-16
> **Response to Reviewer fZuh**
>
> **Q1: The main weakness of this paper comes from the fact that it does not actually compare to all the adequate baselines. So not comparing model-free algorithms at all is a big weakness in my opinion.**
>
> A1: Thank you for your comment. In Section 6.2, we have indeed conducted comparative experiments with a model-free RL algorithm, and we have also provided theoretical analysis regarding the dataset coverage assumption. These findings indicate that a model-free RL is not sufficient for the Offline Extensive-Form (OEF) setting. Since the OEF problem requires establishing a relationship between the dataset and the equilibrium strategy, and since the computation of this equilibrium strategy is a minimax optimization problem, a fully model-free algorithm would be inadequate for solving the equilibrium strategy. As a result, we propose a combination approach that combines elements of both model-based and model-free algorithms to tackle the OEF problem effectively.
>
> **Q2:  So I think the theory needs a full revision to improve rigor and clarity. Right now it is just hand wavy.**
>
> A2: We appreciate your suggestion. The theoretical analysis sections in both Section 5.4 and the Appendix have been revised accordingly. To be more specific, we summarize all our theoretical analyses in Section 5.4. Firstly, we introduce two assumptions for the random dataset and the expert dataset. Then, we provide the performance guarantee for BC and MB methods. Finally, we provide the performance guarantee for our OEF algorithm. In the Appendix section, we also provide the minimal dataset assumption for solving the OEF problem and sample analysis for training the neural network model.
>
> **Q3: How does the error in learning the model of the environment from offline data affect equilibrium finding? Does it need pessimism like most of offline RL? How does BCMB compare to standard model-based learning + CFR / JPSRO + pessimism? If pessimism is already imposed by the model-based algorithm, why do we need BC policy mixing?**
>
> A3:  Thank you for your comments. To analyze how the error in learning the environment mode affects equilibrium finding, we first assume that the offline dataset is the random dataset to eliminate the influence of the dataset. Under our theoretical analysis, the game deduced from the random dataset should be the same as the actual game when there is no error in the learning process. When there is some error in learning the environment model, the trained environment model would be slightly different from the actual game. If the trained environment model causes a change in the rank of the utility, then the equilibrium strategy got from the trained environment model may be very different from the equilibrium strategy for the actual game. In contrast, if the error does not cause a change in the rank of the utility, then the equilibrium strategy got from the trained environment model would be very similar to the equilibrium strategy for the trained environment model. Actually, the environment model is trained to emulate the state transition of the offline dataset, essentially learning the game as represented by this dataset, rather than adopting a pessimism-like approach. As outlined in Section 5.2, when utilizing PSRO or CFR algorithms, these algorithms demand extensive environment-related data. However, the model-based RL algorithm falls short in this aspect since it only provides value information for the state, thus making a combination of model-based RL with PSRO/CFR infeasible. To address this, we propose employing an environment model to substitute the actual environment. Given that the offline dataset is generated by an unknown strategy, the frequency of state-action pairs will vary across different offline datasets. As a result, the trained environment models will also differ and may not accurately reflect the actual environment. Furthermore, the corresponding equilibrium strategies for these trained environment models will not align with the equilibrium strategy for the actual game. To mitigate these discrepancies, we suggest incorporating the behavior cloning policy to resolve the OEF problem.

---

> > ### Comment · Reviewer_fZuh · 2023-07-06
> > **Response to Authors**
> >
> > Thanks so much for responding to my queries. I think my concerns regarding the presentation of theory (rigorousness and formalization in Theorems 5.2 & 5.3) is not resolved. Likewise the answer to Q.3 does not fully address my question -- pessimism is also required when the dataset comes from a random policy, especially in settings when the number of samples is small. Finally, to address the concern regarding baselines, in my opinion, it would be required to actually try out many other methods like TD3+BC, IQL, CQL, etc, which have been shown to perform better in practice on standard benchmarks.

---

### Author Response · Authors · 2023-06-16
**To all the reviewers**

We are grateful for your valuable comments and constructive feedback. Here's a summary of the updates made in the revised PDF document, highlighted in green:

1. Distinguish the differences between our work and previous studies (Introduction section).
2. Include the formal definition of the (Coarse) Correlated Equilibrium (Preliminaries section).
3. Propose several offline and online methods for selecting the combination parameter in our OEF algorithm (Algorithms for Offline Equilibrium Finding section).
4. Revised theoretical results (Algorithms for Offline Equilibrium Finding section) and add sample analysis (Appendix D.2).

Once again, thank you for your insightful comments and feedback.

---

### Decision · Action_Editors · 2023-07-15

**Recommendation:** Reject

**Comment:**

fZuh: "not convinced with the theoretical analysis -- I still find it too informal and it does not discuss a number of steps required for a formal proof. I also think the empirical claims need to be strengthened with more baseline approaches, including more recent approaches"

Patg: "As the authors have explained, I agree it is still a valuable contribution to have a perhaps incomplete, but at least existing, method, that can serve as a baseline for future improvements.
I still have a reserve on the clarity of the theoretical statement, which I think still require some work to be more accessible / more rigorous, thus I stick with "leaning for accept" and not "accept"."

FpMY: "this seems to be the first effort in offline learning for extensive-form games, which is interesting....
I believe the quality of the theoretical statements is insufficient, an opinion that appear to be shared by all the reviewers"

**Audience:**

If accepted this paper would be of interested to the offline and game theory communities.

**Claims And Evidence:**

The paper addresses offline learning for the interactive two-player games setting. The main idea is to learn an equilibrium from prior data, which is then used to for new model based learning algorithms.

All the reviewers agree that the problem is interesting, and a not-well studied. They appreciate the authors responses and clarification of the novelty. At the same time, there is a consensus that the paper in its current form lacks the theoretical and empirical rigor to be suitable for the publication.

**Resubmission Of Major Revision:**

The authors may consider submitting a major revision at a later time.